# Surface factors controlling the volume of accumulated Labrador Sea Water

Yavor Kostov[1,2], Marie-José Messias[1], Herlé Mercier[3], David P. Marshall[4], Helen L. Johnson[5]

[1]U. of Exeter, Department of Geography, Exeter, United Kingdom
[2]British Antarctic Survey, Cambridge, United Kingdom
[3]U. of Brest, Laboratoire d'Océanographie Physique et Spatiale, CNRS, Brest, France
[4]U. of Oxford, Department of Physics, Oxford, United Kingdom
[5]U. of Oxford, Department of Earth Sciences, Oxford, United Kingdom

*Correspondence to*: Yavor Kostov (yastov@bas.ac.uk)

**Abstract.** We explore historical variability in the volume of Labrador Sea Water (LSW) using ECCO, an ocean state estimate configuration of the Massachusetts Institute of Technology general circulation model (MITgcm). The model's adjoint, a linearization of the MITgcm, is set up to output the lagged sensitivity of the watermass volume to surface boundary conditions. This allows us to reconstruct the evolution of LSW volume over recent decades using historical surface wind stress, heat, and freshwater fluxes. Each of these boundary conditions contributes significantly to the LSW variability that we recover, but these impacts are associated with different geographical fingerprints and arise over a range of time lags. We show that the volume of LSW accumulated in the Labrador Sea exhibits a delayed response to surface wind stress and buoyancy forcing outside the convective interior of the Labrador Sea, at important locations in the North Atlantic Ocean. In particular, patterns of wind and surface density anomalies can act as a "traffic controller" and regulate the North Atlantic Current's (NAC) transport of warm and saline subtropical water masses that are precursors for the formation of LSW. This propensity for a delayed response of LSW to remote forcing allows us to predict a limited yet substantial and significant fraction of LSW variability at least a year into the future. Our analysis also enables us to attribute LSW variability to different boundary conditions and to gain insight into the major mechanisms that contribute to volume anomalies in this deep watermass. We point out the important role of key processes that promote the formation of LSW both in the Irminger and Labrador Seas: buoyancy loss and preconditioning along the NAC pathway, in the Iceland Basin, the Irminger Sea, and the Nordic Seas.

## 1. Introduction

Labrador Sea Water (LSW) is a deep watermass that forms during episodes of intense wintertime convection in the Labrador and Irminger Seas (e.g., Pickart et al., 2003a). LSW plays an important role in the decadal and multidecadal variability of the Atlantic Meridional Overturning Circulation (AMOC) (Yeager et al, 2021; Oldenburg et al. 2021), while the connection between LSW and AMOC variability on shorter timescales is unclear (Li et al, 2019). In addition, LSW contributes
significantly to the ocean uptake and storage of heat (Desbruyeres et al., 2014; Messias and Mercier, 2022) and tracers such as carbon and oxygen (Terenzi et al., 2007; Perez et al., 2013; Rhein et al., 2017; Koelling et al. 2022), which can affect the

pace of regional and global climate change. LSW is also rich in dissolved chlorofluorocarbons (CFCs), whose concentration can be used to track the formation (LeBel et al., 2008; Böning et al., 2003), as well as the advection and mixing (Sy et al., 1997; Fine et al., 2002; Rhein et al., 2002; Rhein et al., 2015  Kieke and Yashayaev, 2015) of this water mass away from the Labrador Sea.

The volume budget of LSW in the Labrador Sea is a complex balance between multiple mechanisms that are at play throughout different seasons. In the winter, deep convection in the subpolar North Atlantic depends on a set of prerequisites that pre-condition vertical instability of the water column in the Labrador or Irminger Seas (Pickart et al. 2003a). One of these necessary conditions is the availability of weakly-stratified Subpolar Mode Water, which forms in the eastern subpolar gyre and is a precursor that can get transformed into LSW (Petit et al 2020; Brambilla et al. 2008; McCartney and Talley, 1982).

Under favorable conditions, wintertime heat loss in the Labrador Sea triggers deep convection in the basin and the formation of LSW as a low-stratified water mass (Holdsworth and Myers, 2015; Jung et al., 2014; Schulze et al., 2016). However, the  rate of seasonal surface cooling and watermass formation is vulnerable to anthropogenic climate change due to both projected changes in surface heat fluxes and changes to the meltwater input from Greenland (Garcia-Quintana et al., 2019). In addition, LSW formation rates respond strongly to natural variability. As far back as 1996, Dickson et al. (1996) propose that the North Atlantic Oscillation (NAO), a major mode of atmospheric variability, affects the rate of deep convection in the Labrador Sea. Namely, a positive NAO phase is associated with stronger winter heat loss and a tendency for more enhanced production of LSW. These differences in watermass formation are reflected in the estimated regional uptake of anthropogenic carbon (Rhein et al., 2017). The latter is strong during high NAO periods such as the early 1990s and weak during the late 1990s, which were marked by weaker convection in the Labrador Sea (Steinfeld et al., 2009; Pérez et al., 2010).

The relationship between the NAO and LSW formation can be modulated by other strong factors such as the release of freshwater from the Arctic (Curry et al., 1998; Gerdes et al., 2005; Haine et al., 2008), which inhibits deep convective mixing. Warm-core eddies that shed from the basin boundary near Cape Desolation, the so-called Irminger Rings, are also capable of suppressing deep convection (Czeschel, 2004; Gou et al., 2023). The instabilities that lead to the formation of Irminger Rings are strongest in the winter and weakest in the summer (Gou et al., 2023). Wind stress, too, affects the formation of LSW via multiple mechanisms. Local wind stress can lead to an increase in the eddy kinetic energy near Cape Desolation, which in turn suppresses LSW formation (Czeschel, 2004). In addition, winds along the Greenland coast can drive an Ekman transport of low-salinity water from the boundary towards the convective interior of the Labrador Sea, once again reducing the rate of LSW formation (Czeschel, 2004).

In models and observations, a fraction of the LSW produced during wintertime convective events in the Labrador Sea is exported to the subtropical gyre both along the Deep Western Boundary Current and along interior pathways (Lozier et al., 2012; Rhein et al., 2017) while the remainder recirculates within the subpolar gyre. The relationship between LSW production and export is complex (Zou and Lozier, 2016) because the subpolar gyre stores a large volume of LSW formed over a range of years (Zou and Lozier, 2016). LSW recirculates between the Labrador Sea, the Iceland Basin, and the Irminger Sea (Yashayaev et al., 2007a). In addition, some LSW is formed in the Irminger Sea (Pickart et al. 2003a; Pickart et al. 2003b).

Export and recirculation are not the only processes that deplete the volume of LSW in the Labrador Sea. Once LSW is isolated from the ocean's surface mixed layer, it experiences restratification and mixes with other water masses via isopycnal stirring (Lazier et al., 2002; Yashayaev 2007b). Various processes contribute to this isolation from the surface and subsequent depletion. For example, Irminger Rings restratify LSW (Hátún et al., 2007; Gelderloos et al., 2011; de Jong et al., 2014).

Convection itself generates baroclinic instability, which gives rise to cold core convective eddies (Marshall & Schott, 1999). The latter may restratify the upper portion of the water column and hence isolate LSW from the surface (Marshall & Schott, 1999). Analogous instabilities and restratifying eddies arise from the boundary currents around the whole Labrador Sea (Spall, 2004; Gou et al., 2023). Khatiwala and Visbeck (2000) estimate that boundary current eddies play a significant role in facilitating the so-called "flushing" of LSW out of the basin. However, general circulation models may not represent these

diverse processes and their seasonality consistently and correctly, especially the role of eddies (Gou et al., 2023). Some models tend to overestimate the seasonal depletion of LSW (Li et al., 2019). Hence, models generally underestimate the fraction of LSW stored in the Labrador Sea from one year to another (Li et al., 2019).

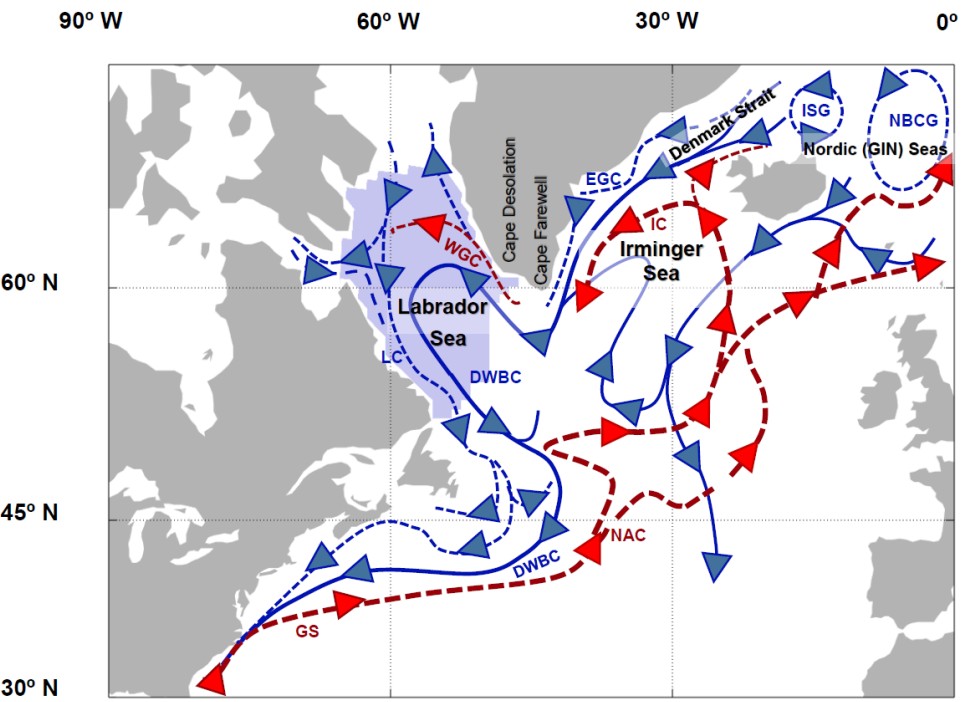

**Figure 1: Spatial mask (light purple shading) of the Labrador Sea region where we calculate the volume of stored LSW. The region**
**is bounded by the OSNAP-West array in the southeast (Lozier et al., 2017). The continental base mask is generated using the free and publicly available software "M_Map: A mapping package for MATLAB", provided by R. Pawlowicz. Superimposed is a schematic of the high-latitude circulation in the North Atlantic ocean following Våge et al. (2013), Houpert et al. (2018), Raj et al. (2019), Florindo-López et al. (2020), and Jutras et al. (2023). Dashed curves represent surface currents, and solid curves deep currents with the direction indicated by arrows. Blue and red denote the transport of relatively cold and warm water, respectively. The thicker curves correspond to relatively larger volume transport. Acronym labels denote: GS (Gulf Stream); DWBC (Deep**
**Western Boundary Current); NAC (North Atlantic Current); LC (Labrador Current); WGC (West Greenland Current); IC (Irminger Current); EGC (East Greenland Current); ISG (Iceland Sea Gyre) ; NBCG (Norwegian Basin Cyclonic Gyre circulation).**

There are various approaches for estimating watermass budgets using data from observations or reanalysis products, and some of these methods have been successfully applied to LSW. Straneo et al. (2003) use Lagrangian float data to study the advection and diffusion of this watermass. Mackay et al. (2020) employ the Regional Thermohaline Inverse Method (Mackay et al., 2018) to estimate the formation, export, and mixing of LSW. Li et al. (2019) use observational Argo float data to calculate LSW volume over the Labrador Sea. Other studies rely on section-based data for LSW layer thickness to estimate watermass volume changes over the whole basin (Yashayaev and Loder, 2009; Rhein et al., 2017). Here we present a different method whose advantage is that we use only surface boundary conditions to reconstruct LSW volume variability. In addition, our approach allows us to obtain limited predictability of LSW volume anomalies a year into the future.

In our method, we use the adjoint of the Massachusetts Institute of Technology general circulation model (MITgcm), a linearization of the model, to estimate the lagged linear sensitivity of LSW volume in the Labrador Sea to surface wind stress, as well as surface fluxes of heat and freshwater. We apply this linearization technique to the ECCO version 4 (ECCOv4) state estimate configuration. We treat the surface boundary conditions in such a way as to avoid overlap and double-counting between their interrelated contributions. For instance, when we analyze the impact of surface winds, we account for their input of momentum. In contrast, the winds' impact on air-sea heat exchange is considered to be part of the heat flux contribution to LSW variability. Similarly, when we talk about the effect of surface heat fluxes, we do not include their impact on surface salinity via evaporation, as that is accounted for in the contribution of surface freshwater fluxes.

Our approach is different from the watermass formation and transformation framework of Walin (1982), Speer and Tziperman (1992), and Desbruyeres et al. (2019) who also use surface fluxes in their analysis. Our main constraint for defining LSW is based on vertical stratification (a component of the potential vorticity, PV), while we also define generous potential density bounds to help identify the watermass. This is similar to the definition of LSW used in Li et al. (2019). Another important difference between our framework and Speer and Tziperman (1992) is that we consider the immediate and delayed impacts of both local and remote surface heat and freshwater fluxes, as well as surface wind stress across the entire Atlantic – Arctic region.

A number of previous studies have applied the adjoint of the MITgcm to exploring sources of ocean variability (e.g., Czeschel et al., 2010; Heimbach et al, 2011; Pillar et al., 2016; Jones et al., 2018; Smith and Heimbach, 2019; Loose et al., 2020; Kostov et al. 2021; Boland et al. 2021; Kostov et al., 2022), but we are the first to use this framework for reconstructing variability in the volume of LSW accumulated in the Labrador Sea. In addition, our method allows us to attribute historical watermass anomalies to different surface boundary conditions and to identify some of the physical mechanisms that govern the LSW volume budget in the ECCOv4 state estimate (Forget et al., 2015; Fukumori et al., 2017; Heimbach et al., 2019). Our results suggest that the upper limb of the AMOC exerts a strong lagged influence on the rate of LSW formation, a feature seen in some but not all general circulation models (Ortega et al., 2017; Li et al., 2019). It is noteworthy and novel that we identify this causal relationship between the upper AMOC limb and LSW in the ECCO state estimate constrained with historical surface boundary conditions and observations of the real ocean (Forget et al., 2015).

In Section 2 we describe our methods for estimating the watermass volume budget using surface boundary conditions. In Section 3, we analyze our results and compare them to observations. In Section 4, we discuss the wider implications of our findings and their relevance to physical processes in the North Atlantic subpolar gyre.

## 2. Methods

### 2.1 General circulation model and its algorithmic differentiation

The state-of-the-art Massachusetts Institute of Technology general circulation model (MITgcm) has been successfully applied in many studies exploring ocean dynamics on regional and global scales (Marshall et al. 1997). Here we use ECCO, a special configuration of the MITgcm: a global ocean state estimate, in which the model is fit to available observations in a least squared sense (Forget et al. 2015). The global state estimate has a nominal 1° horizontal resolution and 50 vertical levels. Heimbach et al. (2019) describe ECCO as a "dynamical interpolator" filling the gaps between heterogeneous observations in a property-conserving and physically consistent fashion. The ECCO estimation framework (Forget et al. 2015; Fukumori et al., 2017; Heimbach et al., 2019) assimilates surface altimetry and ocean bottom pressure data from satellite measurements, sea surface temperature from passive microwave radiometry, sea surface salinity fields from the NASA Aquarius mission (Vinogradova et al., 2014), as well as sea ice concentration fields. Data for the interior of the ocean also comes from Argo floats (Roemmich et al., 2009; Riser et al., 2016), conductivity-temperature-depth sensors, expendable bathythermographs, ice-tethered profiles, tagged seals (Roquet et al., 2013) as well as temperature and salinity climatology from the World Ocean Atlas 2009 (WOA09, Antonov et al., 2010; Locarnini et al., 2010).

The three releases of the global ECCO version 4 (ECCOv4) configuration considered here differ among each other in their periods of integration, the observational data constraints used, and the application of updated numerical schemes in the MITgcm (ECCO Consortium, 2021). ECCOv4 release 2 (ECCOv4r2) spans the time period 1992-2011. The more recent release ECCOv4r3 is integrated between 1992 and 2015 and introduces new additional data constraints: sea surface salinity from the Aquarius mission, ocean bottom pressure from GRACE satellite measurements, and hydrographic profiles in the Arctic ocean (ECCO Consortium, 2021). The ECCOv4r3 controls also include initial conditions for velocity and sea level. Compared to ECCOv4r2, ECCOv4r3 uses modified observational constraints such that the time-mean fields and the anomalies are treated separately (ECCO Consortium, 2021). However, in release 4 (ECCOv4r4), which covers the 1992-2017 period, time-mean and time-varying controls are no longer separated as in ECCOv4r3. Another important difference between ECCOv4r4 and earlier releases is the use of updated and more stable numerical schemes in the MITgcm (ECCO Consortium, 2021).

The ECCOv4 configuration reproduces very well the observed time-mean and variability of the Atlantic meridional overturning circulation, including transport across the RAPID (Jackson et al., 2019; Kostov et al., 2021), OSNAP, and OVIDE arrays (Kostov et al., 2021). ECCOv4 also exhibits a realistic density structure in the Labrador Sea (Jackson et al., 2019). However, ECCO may be overemphasizing the role of salinity for setting this structure during recent historical periods (Jackson et al., 2019).

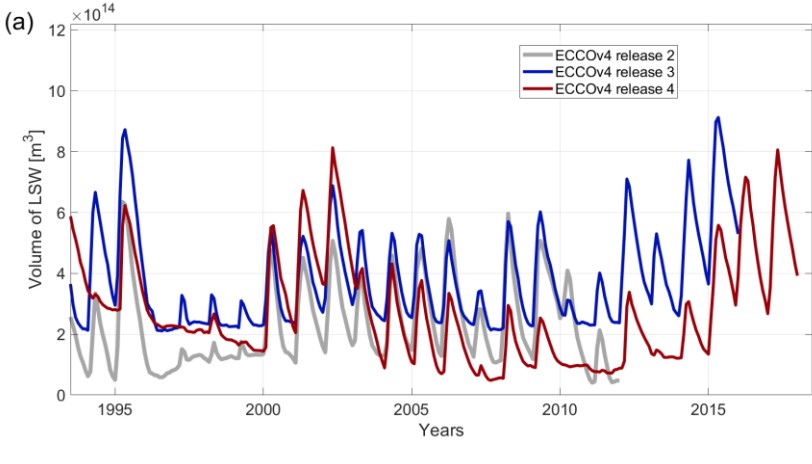

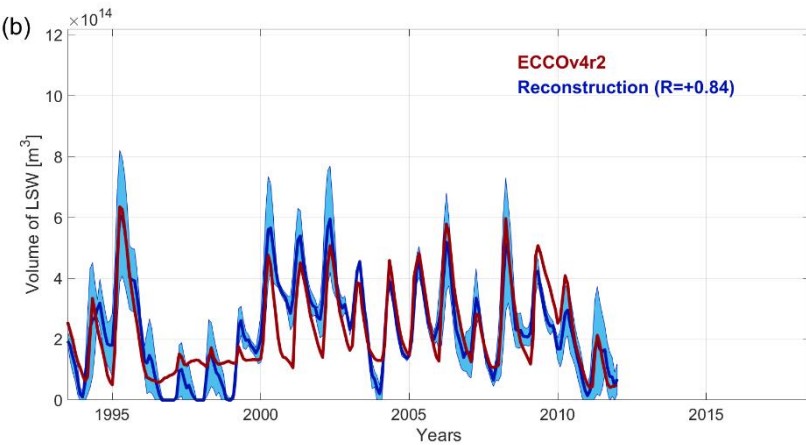

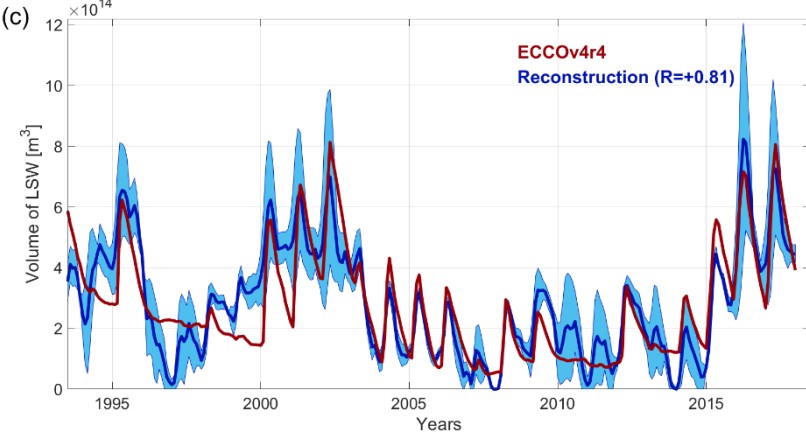


**Figure 2: (a) Monthly-mean timeseries of LSW volume [$m^3$] in the ECCOv4r2 (gray), ECCOv4r3 (blue), and ECCOv4r4 (red) state estimate based on the definition in Sec. 2.2; (b) Monthly-mean timeseries of LSW volume [$m^3$] in the ECCOv4r2 state estimate (red) and in our monthly-average reconstruction (dark blue line, Eq. 7 and Sec. 3), which includes recovered monthly anomalies and the ECCO climatological seasonal cycle. The reconstructed LSW volume is set to be strictly nonnegative at the end of 1996, 1997, and**

**1999. Blue shading reflects the uncertainty in the mean estimate approximated as the difference between the minimum and the maximum of the three different reconstructions at each point in time; (c) same as in b but for the ECCOv4r4 state estimate (red) and our reconstruction (dark blue curve) with indicated uncertainty (shaded blue envelope).**

The time-mean, the seasonal cycle, and the anomalies in LSW volume in the Labrador Sea (Fig. 1) of the ECCOv4 state estimate releases (Forget et al., 2015), used in this study, are on the order of $10^{14}$ $m^3$ (Fig. 2). This is similar to basin-wide
observational estimates of $3.37 \times 10^{14}$ $m^3$ for the 2003-2007 winter volume maximum (Li et al., 2019). The annual mean volume of LSW and the anomalies relative to the seasonal mean in ECCO are on the same order of magnitude, $10^{14}$ $m^3$, which gives rise to a very fluctuating timeseries of historical watermass variability (Fig. 2). The large historical variability in LSW that ECCOv4 exhibits (across ECCOv4r2, ECCOv4r3, and ECCOv4r4 releases) is realistic and consistently reproduces certain well-documented anomalies in the watermass volume: the deep convection (and thicker layer of LSW) in the early 1990s, the reduced
depth of convection and LSW volume in the late 1990s (Rhein et al., 2017), and the partial recovery of LSW formation after 2015 (Yashayaev and Loder, 2017; Yashayaev et al., 2023). The ECCO state estimate also reproduces the 2008 rise in watermass volume (Yashayaev and Loder, 2009), although this increase in 2008 is less pronounced in ECCOv4r4, compared to ECCOv4r3.

One of the most advanced features of the MITgcm is that the model code is automatically differentiable (Giering, 2010), which allows for the generation of an adjoint linearization (Heimbach et al., 2011; Fukumori et al., 2017; Marotzke et al, 1999;
Forget et al., 2015). This capability is essential for the development of the ECCO ocean state estimate (Fukumori et al., 2017; Forget et al., 2015). The adjoint of the model outputs the linear sensitivity (first derivatives) of a specified ocean index, an objective function, with respect to parameter choices and different boundary conditions over a range of lead times.

## 2.2 Objective Function

In this work, our objective function of interest is the volume of LSW in the Labrador Sea region (Fig. 1), bounded by the
OSNAP-West line (Lozier et al., 2017). Following Talley and McCartney (1982), Zou and Lozier (2016) and Li et al. (2019), we define LSW using a combination of two criteria. The first criterion is based on potential vorticity (PV),

$$PV < PV_{upper} \tag{1}$$

where the upper bound is $PV_{upper} = 4 \times 10^{-12}$ $m^{-1}s^{-1}$ as in Zou and Lozier (2016). Following Zou and Lozier (2016) and Li et al. (2019), we approximate PV in terms of the vertical stratification:

$$PV \approx f\frac{N^2}{g} \tag{2}$$

where $f$ is the Coriolis parameter, $g$ is the gravitational acceleration, and $N$ is the buoyancy frequency. Thus, the PV condition which we impose ensures that we define LSW as a weakly stratified watermass. Li et al. (2019) demonstrate that in observations and in most models, this criterion is universal and sufficient for identifying LSW in the Labrador Sea. However, in some models and model configurations, low stratified water can be found below the LSW layer in the basin (Li et al., 2019).
Thus, we introduce a second constraint, which sets bounds on the potential density $\sigma_\theta$ of the watermass referenced to the surface:

$$\sigma_{\theta \, lower} < \sigma_\theta < \sigma_{\theta \, upper} \tag{3}$$

where $\sigma_{\theta \, lower} = 27.7\,\mathrm{kg\,m^{-3}}$ and $\sigma_{\theta \, upper} = 27.84\ \mathrm{kg\,m^{-3}}$. This potential density constraint is deliberately very generous because in ECCO, similarly to many models and observations, the density, temperature, and salinity of the LSW formed in the

Labrador Sea differ from year to year (Feucher et al., 2019). However, we have tested a stricter density constraint with $\sigma_{\theta \, upper} = 27.81\ \mathrm{kg\,m^{-3}}$. Using $27.81\ \mathrm{kg\,m^{-3}}$ as the upper bound gives the same results for the volume of LSW in ECCOv4r2. In contrast, in the subsequent ECCOv4r3, the Labrador Sea in ECCO has a secondary deep layer of low-stratified water denser than $27.81\ \mathrm{kg\,m^{-3}}$ and distinct from the LSW above. The existence of this deep low-stratified water explains most of the time-mean offset between our calculation of LSW volume in ECCOv4r3 compared to releases 2 and 4. In the most

recent ECCOv4r4, there is only a brief period between the mid to late 1990s where the model produces and then stores low-stratified water all the way down to $27.84\ \mathrm{kg\,m^{-3}}$ in the Labrador Sea. Outside this period both $\sigma_{\theta \, upper} = 27.84\ \mathrm{kg\,m^{-3}}$ and $\sigma_{\theta \, upper} = 27.81\ \mathrm{kg\,m^{-3}}$ give the same results for the volume of LSW in ECCOv4r4.

We formulate our objective function for the volume of LSW in the Labrador Sea ($Vol$) such that it is continuously differentiable by the algorithmic differentiation software (Giering, 2010). We therefore impose the PV and potential density

criteria (Eq. (1) and (3)) by using logistic activation functions $A(a,b)$ of the form

$$A(a,b) = (1/2 + \tanh((a-b)\,c)/2) \tag{4}$$

that approximate the Boolean conditions of the form $a > b$ in our definition of LSW (e.g., Fig. A1 in Appendix A). Our objective function is thus defined as:

$$
\begin{aligned}
Vol \;=\; & \frac{1}{(t_{end} - t_0)} \sum_{t_0}^{t_{end}} \sum_{x,y,z} GridCellVolume(x,y,z) \\
& \cdot A(\sigma_\theta(x,y,z,t), \sigma_{theta\ lower}) \\
& \cdot \left(1 - A\big(\sigma_\theta(x,y,z,t), \sigma_{theta\ upper}\big)\right) \\
& \cdot \left(1 - A\big(PV(x,y,z,t), PV_{upper}\big)\right) \\
& \cdot A(PV(x,y,z,t), -1000)
\end{aligned}
\tag{5}
$$

where $c = 10^5\ kg^{-1}\,m^3$ or $c = 10^5\ m\,s$ is a factor that controls the slope of the activation function with respect to the input, and $GridCellVolume$ is the volume of each grid cell. The last factor in our formula represents a lower bound on $PV > -1000 \times 10^{-12} m^{-1} s^{-1}$, which in practice is always satisfied in our case and includes vertically unstable water columns in the Labrador Sea. For a discussion of the activation functions, see Appendix A and Figs. A1 and A2. The right hand side of Eq. (5) is summed over the model's horizontal $x, y$ and depth ($z$) coordinates within the Labrador Sea region bounded by the

OSNAP-West line (Lozier et al., 2017) to the southeast (Fig. 1). The objective function is furthermore averaged in time over a simulated time interval $t_{end} - t_0$ which is equal to 1 month.

Our objective function, LSW volume, has values on the order of $10^{14}\ m^3$ and exhibits variability on the same order of magnitude. However, we do not compute the objective function in units of $10^{14}\ m^3$. Instead, we multiply the LSW volume by a large arbitrary nondimensional factor of 1500. The rescaling increases the magnitude of the sensitivity patterns output by the adjoint. This eliminates some (but not all) of the numerical noise that arises when the adjoint of the MITgcm outputs the sensitivity of our objective function to surface boundary conditions. The rescaling helps as some sources of numerical noise in the adjoint have a magnitude independent of the magnitude of the objective function and its sensitivity. We then divide the lagged sensitivity patterns that the adjoint outputs by 1500 when we post-process them offline, so that the post-processed sensitivity has units of $m^3$ per forcing units, assuming that the forcing is sustained over a single model time step set to 1 hour.

The value of our objective function and its sensitivity to boundary conditions depend on the season for which we compute the objective function. The sensitivity of wintertime LSW volume to past surface boundary conditions is different from the sensitivity of summertime LSW volume. However, calculating the sensitivity of objective functions evaluated over each calendar month is too computationally expensive. Instead, as in Kostov et al. (2021), we conduct 4 adjoint calculations where the objective function is evaluated over four different representative months of the year, corresponding to spring, summer, fall, and winter seasons during the 2006-2007 historical period of the ECCOv4r2 state estimate. That is, we assume that a monthly objective function from March is representative of January and February monthly objective functions, too. Similarly, an August objective function is assumed to be a good substitute for July and September monthly objective functions. In addition to the 2006-2007 set of seasonally representative objective functions, for comparison, we do adjoint calculations with wintertime objective functions computed in 2006 and 2011, and summertime objective functions computed in 2005 and 2010. The years 2006, 2007, and 2011 represent diverse regimes of the NAO (Fig. B1), which is among the major drivers of ocean variability in the North Atlantic subpolar gyre (Dickson et al. 1996; Rhein et al., 2017; Roussenov et al., 2022). These three years are also marked by different winter mixed layer depths in the Labrador Sea and different volume of LSW in the background ocean state (Fig. B1). The selected three years are close to the end of the ECCOv4r2 state estimate because that allows us to compute lagged sensitivity over longer periods leading up to these three years. We use the sensitivity patterns for the ECCOv4r2 objective functions to reconstruct not only LSW variability in the same ECCOv4r2 release, but also to attempt reconstructing LSW in the more recent ECCOv4r3 (not shown) and in ECCOv4r4, that extends further in time until the end of 2017.

## 2.3 Using lagged sensitivity to reconstruct and attribute variability

When we define the LSW volume $Vol$ as our objective function, we compute its linear sensitivity $S$ with respect to surface boundary conditions. For example, we consider surface heat fluxes $Q$ at geographical location $\boldsymbol{x}$ and at a lead time $t_{lead}$ and compute the sensitivity

$$S(Q, \boldsymbol{x}, \tau) = \frac{\partial Vol}{\partial Q(\boldsymbol{x}, t_{lead})} \tag{6}$$

in units of [$m^3 W^{-1} m^2$ over 1 hour]. For each lead time $t_{lead}$, the adjoint of the model produces spatial maps that show the sensitivity of the LSW volume to hypothetical surface heat fluxes if they were applied over 1 model timestep (equal to 1 hour).

These sensitivity patterns tell us whether warming or cooling each part of the ocean surface causes a delayed increase or decrease in the volume of LSW. Similarly, we obtain analogous sensitivity maps with respect to surface freshwater fluxes $F$, and zonal (positive eastward) and meridional (positive northward) wind stress, $\tau_E$ and $\tau_N$, respectively.

The linear sensitivity patterns generated by the adjoint can be used to reconstruct and attribute variability in the objective function of interest, in our case, LSW volume. We convolve the lagged sensitivity to each surface boundary condition $B$ with

the time-history of anomalies in that boundary condition over a range of lead times $t_{lead}$. We repeat the same approach for heat and freshwater fluxes, as well as surface winds,

$$Vol(t) \approx \sum_{B=Q,F,\tau_N,\tau_E} \int_0^{t_{cutoff}} \int S(B, \boldsymbol{x}, t_{lead}) B(t - t_{lead}) \, d\boldsymbol{x} \, dt_{lead} \tag{7}$$

to recover a timeseries of the anomalies in watermass volume $Vol(t)$. We use this method to estimate variability of LSW relative to the seasonal or annual mean watermass volume. For practical purposes, in our study, we choose a maximum cut-

off lead-time $t_{cutoff} = 6.5 \; years$ beyond which we assume no memory of past surface forcing (See Fig. C1 and the discussion in Appendix C). This choice does not have a large impact on our reconstruction skill. By using sensitivity patterns from wintertime objective functions evaluated in 2006, 2007, and 2011, we obtain three different reconstructions. Looking at the difference between the minimum and the maximum of these reconstructions at each point in time gives us a rough estimate of the uncertainty due to linearizing the model simulation over particular historical periods with different background states of

the ocean.

If surface forcing at lead-times greater than zero contributes significantly to the reconstruction of LSW volume variability, we can use this information to make predictions about future anomalies in $Vol(t)$ over a time horizon $t_{horizon}$:

$$Vol_{prediction}(t) = \sum_{B=Q,F,\tau_N,\tau_E} \int_{t_{horizon}}^{t_{cutoff}} \int S(B, \boldsymbol{x}, t_{lead}) B(t - t_{lead}) \, d\boldsymbol{x} \, dt_{lead} \tag{8}$$

In addition to reconstructing and predicting variability, this approach can also be used for causal attribution. We can

decompose the reconstruction in Eq. (7) into separate contributions from wind stress, heat, and freshwater fluxes at the surface. This allows us to highlight the individual roles of these different sources of variability.

The default sensitivity patterns calculated by the adjoint contain built-in information about air-sea feedback represented using bulk formulae and a parameterization of surface radiation. For example, the default sensitivity patterns assume that anomalous shortwave heat flux into the ocean subsequently triggers a combination of radiative cooling, evaporation, and/or

turbulent heat flux out of the ocean (Kostov et al., 2019). This response of the ocean is also part of the net surface heat flux budget. Therefore, both the net surface heat flux $Q$ and the default sensitivity patterns $S(Q, x, t - t_{lead})$ output by the adjoint include air-sea feedback. As a result, the convolution in Eq. (7) can erroneously double count air-sea feedback mechanisms.

In order to avoid this problem, we cannot rely on the default configuration of the adjoint. Instead, we have to instruct the algorithmic differentiation software not to take derivatives of the bulk formulae and the surface radiation parameterization

code. This approach guarantees that our lagged sensitivity patterns do not include air-sea feedback effects that are already

accounted for in the net surface heat flux budget and the net surface freshwater budget. For example, the effect of surface heat fluxes on surface salinity via evaporation is accounted for only in the surface freshwater budget. In addition, the impact of evaporation and precipitation on surface temperature via latent heat fluxes is accounted for only in the surface heat flux budget. However, following our approach, the model's forward trajectory remains the same as in the optimized ECCO state estimate.

## 2.4 Identification of preferred lead times in the ocean system

We can use our linear convolution framework to gain additional insight into the lead-lag relationships between surface boundary conditions in different regions and the volume of LSW. In other words, we can identify if surface boundary conditions in certain geographical regions have preferred lead times at which they contribute to LSW volume anomalies. For example, we can look at different lead times $t_{lead}$ and compute the contribution that surface heat fluxes $Q$ over a particular region make at each of these lead times:

$$Vol_Q(t, t_{lead}, X) = \int S(Q, x, t_{lead}) Q(t - t_{lead}) X(x) \, dx \tag{9}$$

where $X(x)$ is a regional spatial mask set to 1 in the region of interest and 0 everywhere else.

We consider the Pearson correlation $R$ between $Vol_Q(t, t_{lead}, X)$ for each lead time $t_{lead} = T$ on one hand, and the sum over all lead times $\sum_{t_{lead}} Vol_Q(t, t_{lead}, X)$ for a given region $X$:

$$R(T) = R\left[Vol_Q(t, t_{lead} = T, X), \sum_{t_{lead}} Vol_Q(t, t_{lead}, X)\right] \tag{10}$$

However, the Pearson correlation does not give us an idea about the *magnitude* of variability at each lead time that projects onto the regional contribution of a particular boundary condition to LSW anomalies. To estimate this magnitude in units of $m^3$, we can weight $R(T)$ from Eq. (10) by the standard deviation of $Vol_Q(t, t_{lead} = T, X)$:

$$R(T) \, std\left(Vol_Q(t, t_{lead} = T, X)\right) \tag{11}$$

Peak values of the weighted lagged correlation (Eq. (11)) with respect to $t_{lead}$ indicate characteristic or preferred lead times at which a given geographical region $X$ contributes to LSW volume variability.

## 2.5 Identification of key geographical regions and major processes as sources of remote influence on LSW volume anomalies

In addition to identifying preferred lead times in the system, we can use the sensitivity patterns $S$ generated by the adjoint to identify the most important regions of the ocean that exert a delayed impact on the volume budget of LSW. The patterns $S$ represent the hypothetical proclivity of LSW to respond to surface boundary conditions. However, there has to be actual variability in these surface boundary conditions to activate the sensitivity patterns. Therefore, we are interested in how variability in each surface boundary condition projects on the spatial sensitivity patterns at different lead times. For example, we may consider maps of the standard deviation of net surface heat fluxes $std_Q(x)$ and use these to weight $S(Q, x, t_{lead})$ pointwise at each model grid cell:

$$std_Q(x) \cdot S(Q, x, t_{lead}) \tag{12}$$

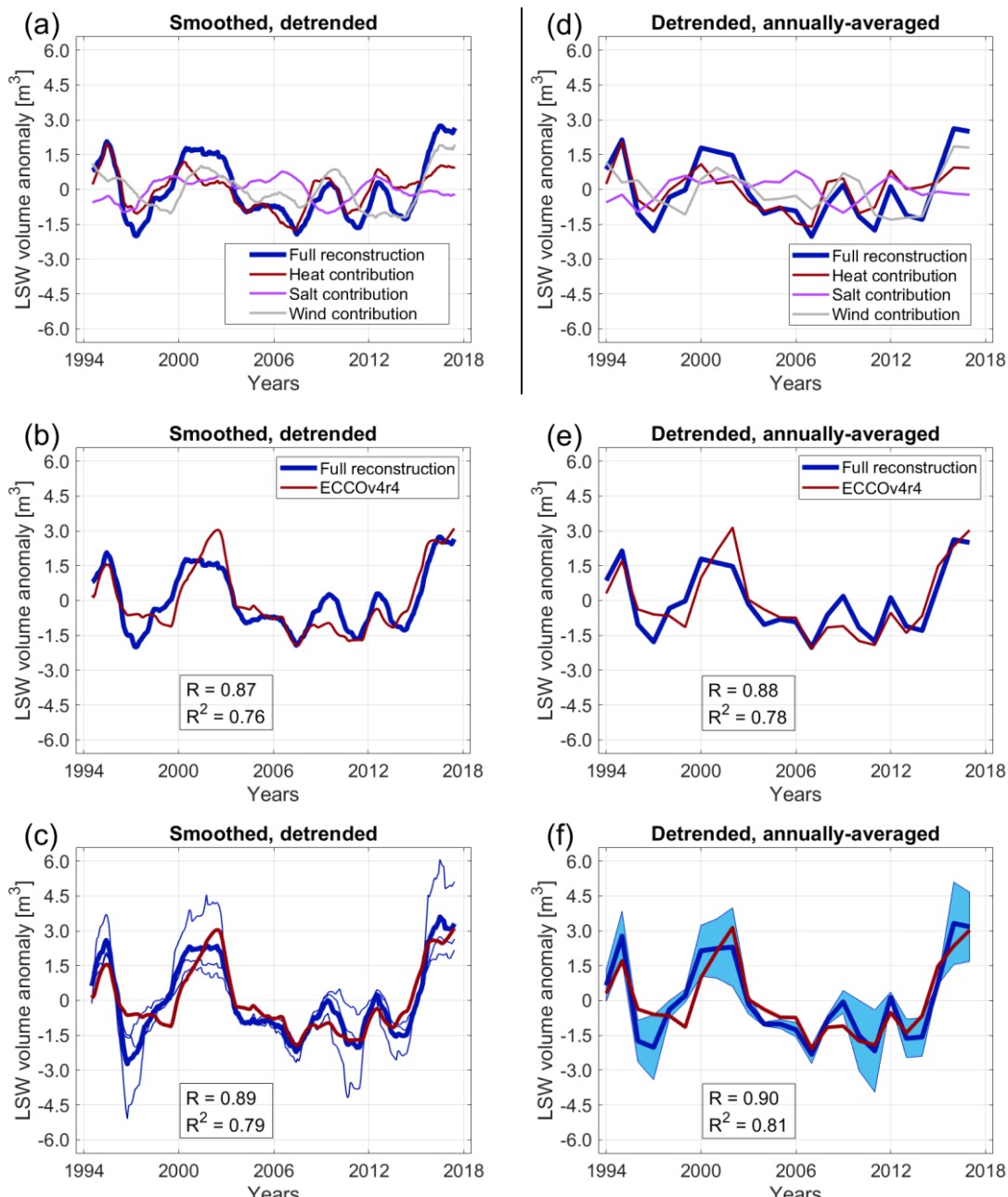

320

**Figure 3:** (a) Contribution of wind stress (both zonal and meridional in gray), heat (red), and freshwater fluxes (purple) at the surface to the monthly reconstruction of ECCOv4r4 LSW volume anomalies (blue) relative to the seasonal cycle. The attribution and reconstruction uses sensitivity patterns from objective functions over the spring 2010 – winter 2011 period in ECCOv4r2 and surface boundary conditions from ECCOv4r4. The timeseries are smoothed with a 13-month running mean. Each contribution

timeseries was detrended. (b) Comparison of the reconstruction (blue) from (a) with the deseasonalized LSW volume anomalies in ECCOv4r4 (red), smoothed with a 13-month running mean. Detrending was applied after the individual contributions were summed ; (c) Same as (b) but also using summertime and wintertime objective functions in 2005-2006 and 2006-2007 to obtain three

different reconstructions (thin blue lines) and a mean estimate (thick blue line). Correlations are shown with respect to the mean reconstruction. (d)-(f): same as (a)-(c) but for annually-averaged contributions and reconstruction timeseries relative to the annual-mean. The shaded envelope in f indicates the difference between the minimum and the maximum of the three reconstructions at each point in time. In (a)-(c) ticks along the horizontal axis mark the beginning of a given year, while in (d)-(f) they denote yearly data points. Pearson correlation coefficients R are indicated on each panel.

These projections highlight the geographical regions that make the largest delayed contributions to LSW variability. For instance, Eq. (12) can identify parts of the ocean where surface heat fluxes tend to trigger LSW volume anomalies at a lead time $t_{lead}$.

In order to understand the physical processes that relate surface boundary conditions to delayed responses in the LSW budget, we combine our adjoint calculations with forward perturbation experiments, as follows. We take the sensitivity to surface freshwater fluxes $F$ at a given lead-time $S(F, \boldsymbol{x}, t_{lead})$ and apply this spatial pattern as a perturbation to the freshwater flux field $F(\boldsymbol{x})$ in the model:

$$F_{perturbed}(\boldsymbol{x}) = F(\boldsymbol{x}) + \mu \cdot S(F, \boldsymbol{x}, t_{lead}) \tag{13}$$

where $\mu$ is a scaling constant. The results of such a perturbation experiment, relative to the unperturbed simulation, reveal what physical mechanisms are activated on different timescales when boundary conditions project onto the sensitivity pattern $S$.

## 3. Results

Using the time-history of surface wind-stress, heat flux, and freshwater flux, along with sensitivity patterns generated by the MITgcm adjoint, we reconstruct monthly-averaged timeseries of the LSW volume anomalies relative to each month of the seasonal cycle (Eq. (7)). We then add our reconstructed anomalies to the climatological seasonal cycle from ECCO and compare the sum with historical LSW variability in the state estimate. Linearizing the model over different time periods allows us to produce three different sets of sensitivity patterns and hence, three different historical reconstructions. This ensemble gives us an envelope of uncertainty which reflects the importance of the background ocean state for the response of LSW to surface boundary conditions (shading in Fig. 2). Note that individual reconstructions generally agree on the sign of the LSW volume anomalies but differ in the estimated magnitude of the anomalies with uncertainty reaching values on the order of $10^{14} \ m^3$. Overall, the magnitude of reconstructed LSW volume variability is consistent with the observational estimates of Li et al. (2019).

Our ensemble-mean results reproduce relatively well the ECCOv4r2 (Pearson correlation coefficient R=+0.84) and ECCOv4r4 (Pearson correlation coefficient R=+0.81) timeseries of LSW volume (Fig. 2), while individual ECCOv4 releases also differ in their representation of historical LSW variability (Fig 2a). Our reconstructions show certain episodic mismatches, e.g., during some of the winter months and over the 1996-1999 period marked by low LSW accumulation in the Labrador Sea. Our reconstruction markedly underestimates the LSW volume in the second half of 1996 and even reaches unrealistic values: we have to impose a separate condition that the total LSW volume in the Labrador Sea is strictly nonnegative. These deficiencies in the reconstruction are likely due to historical changes in the ocean's background state, as well as processes not captured by the model linearization. Our reconstructions (Fig. 2b,c) also exhibit mismatches during some of the seasonal transitions from high to low

LSW volume. It is also important to point out that we only have three-member ensembles of monthly objective functions corresponding to winter and summer times. In contrast, we have a single objective function representing spring months and only one corresponding to the fall. Any small spread between the three reconstructions in the spring and fall is solely due to the subtraction of different time-mean values and the removal of different long-term trends.

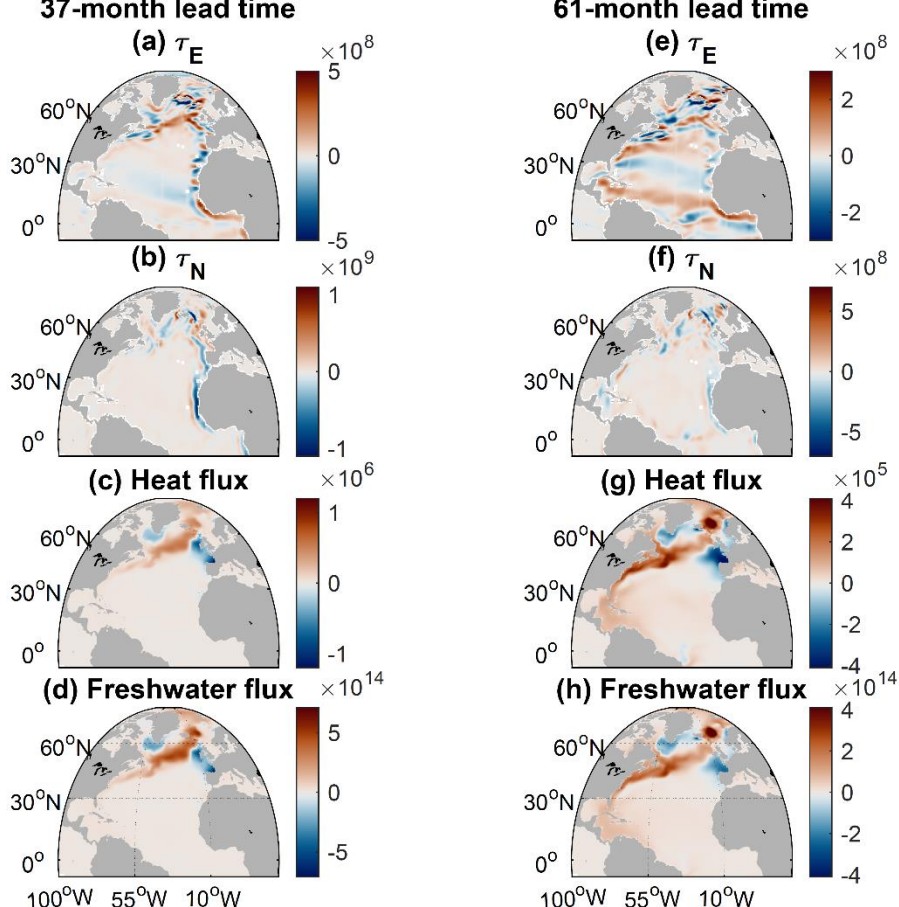

**Figure 4: Sensitivity of wintertime LSW volume in 2007 to surface boundary conditions at a lead time of 37 months ((a) through (d)) and 61 months ((e) through (h)). Sensitivity to zonal (positive eastward) (a,e) and meridional (positive northward) (b,f) windstress $[m^3 N^{-1} m^2$ over 1 hour], surface heat fluxes (c,g) $[m^3 W^{-1} m^2$ over 1 hour] out of the ocean, and surface freshwater fluxes $[m^3 m^{-1} s$ over 1 hour] out of the ocean (d,h). Red shading indicates that a positive anomaly in the surface boundary condition leads to an increase in LSW volume at the indicated lead time.**

We can also compare our result with LSW anomalies relative to the seasonal cycle, where both our reconstruction and the ECCO timeseries are deseasonalized and then smoothed with a 13-month running mean (to mitigate any seasonal bias in our reconstruction). We assume that we need *at least* 24 months of previous history of the surface boundary conditions to reconstruct a given monthly anomaly (Fig. C1). In addition, the 13-month running mean that we apply takes a sliding window of 6 months

before and after each reconstructed month. Hence, in our comparison of the deseasonalized smoothed ECCO timeseries and our reconstruction, we leave out the first 24+6 = 30 months of each timeseries. In this case, we are able to recover more than three quarters of the variability (Pearson correlation coefficient R=+0.89) in LSW volume (Fig. 3c) in ECCOv4r4.

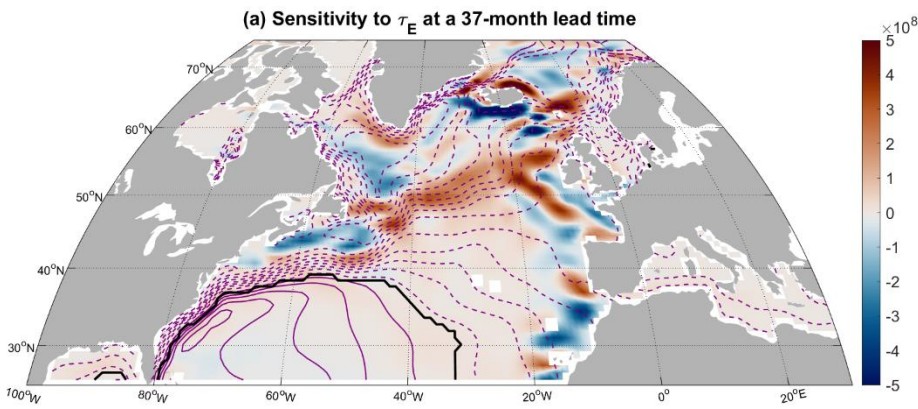

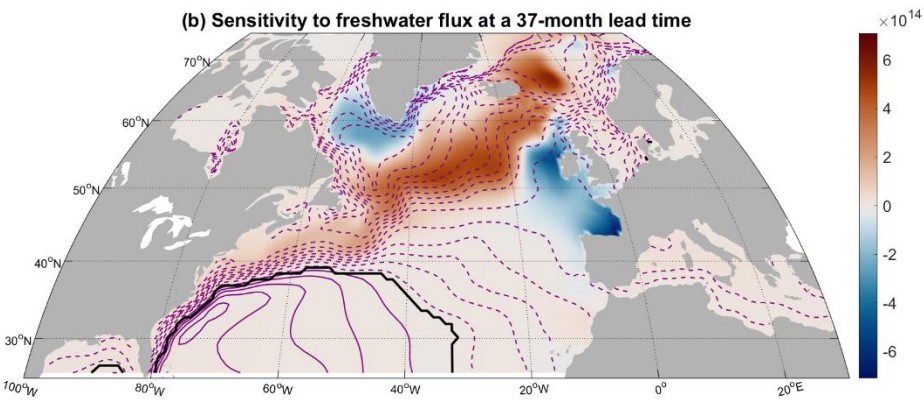


**Figure 5: (a) Same as Fig. 4a [$m^3N^{-1}m^2$ over 1 hour] with windstress defined as positive eastward; and (b) same as Fig. 4d [$m^3m^{-1}s$ over 1 hour] with freshwater flux defined as positive out of the ocean, but focusing on the mid and high latitudes. Superimposed are the climatological time-mean SSH contours from the ECCOv4r2 state estimate, 7 cm apart. Positive contours are solid purple lines, negative contours are dashed, and the zero contour is a thick black line.**

Our method allows us to break down the reconstruction into components due to wind-stress, heat flux, and freshwater flux anomalies at the ocean surface. We see comparable contributions from each of these sources of historical variability (Fig. 3). For example, salt fluxes, heat fluxes, and wind stress, all, contribute to the positive LSW volume anomalies in the early 2000s. We suggest that the 2008 relative increase in LSW explored by Yashayaev and Loder (2009) can be attributed primarily

to surface heat fluxes (Fig. 3a). These attribution results hold true across our three reconstructions. However, all releases of
ECCOv4 seem to underestimate the magnitude of the 2008 relative increase in LSW, where the underestimation is most
pronounced in ECCOv4r4. We find that the subsequent 2010-2011 decline in LSW volume is dominated by heat fluxes and wind
stress, while the 2012 recovery is attributable to both heat and salt fluxes (Fig. 3a). The well-documented increase in LSW volume
after 2015 (Yashayaev and Loder, 2017) seems to be primarily related to wind stress anomalies and, to a smaller extent, surface
heat fluxes.

A similar picture emerges when we consider an annually averaged reconstruction of LSW volume anomalies (Fig. 3f)
relative to the climatological annual mean in ECCO. We once again assume a minimum required time history spanning two years
of surface forcing and thus start our comparison in 1994 rather than 1992, the first year of ECCO. In this case, we recover 81%
of the ECCOv4r4 interannual variability (Pearson correlation coefficient R=+0.90) in LSW volume. Once again, each of the
different surface boundary conditions (wind stress, heat, and salt fluxes) make comparable and equally important contributions
(Fig. 3d).

We now move on to identify geographical regions where the lagged LSW sensitivity to surface boundary conditions is
most pronounced at different lead times. At lead times shorter than a year, the watermass volume is most sensitive to changes
in the Labrador Sea itself such as local heat or freshwater flux anomalies (Fig. D1c,d). The sensitivity to zonal winds at lead
times shorter than a year appears spatially noisy (Fig. D1a).

We then consider sensitivities on interannual timescales (e.g., 3 and 5 years). At lead times of several years, we see
sensitivity to wind stress forcing along the West European and Northwest African shelf, especially in the meridional component
of wind stress (Fig. 4b,f). Coastal waveguides allow the wind-driven upwelling or downwelling signal to be transmitted to the
subpolar gyre, where it alters the sea-surface height (SSH) and density gradients and affects geostrophic transport in the North-
eastern Atlantic, as suggested by Pillar et al. (2016), Jones et al. (2018), Loose et al. (2020), and Kostov et al. (2021). We also
see sensitivity to zonal wind stress along the boundaries of the Labrador Sea (Figs. 4a, 5a), which can affect the strength of the
boundary current and its exchange with the interior (Czeschel, 2004).We further notice that at a lead time of three years, LSW
volume is sensitive to zonal wind stress just south of Greenland (Figs. 4a, 5a) in the region of the Greeland Tip Jet, where
zonal winds can affect the rate of deep convection in the Irminger Sea (Pickart et al., 2003b).

Another important feature in the sensitivity to wind stress is the pattern around the coast of Iceland. We see that a
westward wind stress anomaly along the south coast and an eastward wind stress anomaly along the northern coast of Iceland
promote a larger LSW volume at a lead time of three years (Figs. 4a, 5a). Jones et al. (2018) identify a similar pattern in the
sensitivity of Labrador Sea heat content to wind stress. In addition, Loose et al. (2020) see this pattern in the sensitivity of heat
transport across the Iceland-Scotland Ridge to wind stress. Loose et al. (2020) highlight Ekman transport along the Icelandic
Coast as the mechanism behind this pattern and argue that it can generate onshore convergence or divergence and hence a
pressure anomaly along the coast. This pressure anomaly is quickly communicated around Iceland as a coastal wave and affects
the geostrophic transport between Iceland and Scotland, heat transport convergence in the Nordic Seas (Loose et al., 2020),

and subsequent water mass transformation. Through the Denmark Strait (Fig. 1), temperature and salinity anomalies from the Nordic Seas, are then exported back to the eastern subpolar gyre, where they can precondition LSW formation.

At lead times of several years, we also see remote sensitivity to zonal wind stress, but at these lead times the geographical pattern is mostly aligned with the mean pathway of the North Atlantic Current (NAC) branches. This sensitivity along the NAC is well illustrated in Fig. 4a,e and Fig. 5a, where we use the climatology of SSH contours to indicate the location and direction of the time-mean currents in the subpolar gyre, which is largely wind-driven. Hakkinen and Rhines (2009) suggest that the pathways of the NAC branches have exhibited historical shifts affecting the transport of subtropical water to the Nordic Seas between 1991 and 2005. Significant variability in the NAC flow that branches towards the Nordic Seas has also been observed over the more recent 2008-2016 historical period (Weijer et al. 2022). Raj et al. (2018) and Holliday et al. (2020) suggest that leading modes of atmospheric variability such as the NAO can trigger intensification or weakening of the Norwegian Current which branches from the NAC and transports warm water towards the Greenland-Iceland-Norwegian (GIN) Seas.

At multiannual lead times, we also see sensitivity to surface heat and freshwater fluxes in the GIN Seas (Fig. 4c,g,d,h and Fig. 5b), especially in the region of the cyclonic gyre circulation in the Norwegian Basin (Fig. 1 and Raj et al., 2019). We also see very pronounced lagged sensitivity of LSW volume to surface heat and freshwater fluxes (Fig. 4c,g,d,h and Fig. 5b) both along the NAC and along its flanks at multiannual lead times. We call these distinctive sensitivity patterns a "Traffic Controller" – we propose that input of momentum or buoyancy with this spatial fingerprint can act to accelerate or decelerate the NAC and deflect it away or towards the Irminger Sea, the Iceland Basin, and the Nordic Seas. This in turn affects the transport of warm, saline subtropical water that subsequently undergoes buoyancy loss in the Iceland and Irminger basins (Petit et al., 2020; Petit et al., 2021) and eventually gets transformed into deeper watermasses such as LSW.

We explore our "Traffic Controller" hypothesis in a forward experiment and analyze the mechanisms behind the sensitivity patterns in Fig. 4, including the potential role of the GIN seas. The spatial pattern of the sensitivity to surface freshwater fluxes is very similar to the sensitivity to heat fluxes (Fig. 4). However, compared to heat content anomalies, freshwater anomalies do not directly trigger air-sea feedback mechanisms that are excluded from our sensitivity patterns. We thus choose to apply the sensitivity pattern from Fig. 4d as a perturbation to the background rainfall throughout January 2000 of the ECCO historical state estimate over an extended North Atlantic region (north of 20°N, west of 20°E, south of Fram Strait, including marginal seas but excluding the Mediterranean and the Baltic). However, we multiply the pattern by a factor of $(-10^{-22})$, so that the rescaled perturbation is of order $10^{-8} \ m \ s^{-1}$ (or $\sim 10^{-5} \ kg \ m^{-2} \ s^{-1}$), which is comparable to the standard deviation in January surface freshwater fluxes between different years (see Fig. 8b). Perturbations applied during the winter are distributed over a deeper mixed layer, which further enhances their persistence and triggers a large response. This motivates our choice to branch the experiment from the ECCO state estimate (our control run) in January. On the other hand, the perturbation we prescribe is based on a sensitivity pattern from a winter-time objective function, so it would not be appropriate to extend the prescribed forcing anomaly beyond the winter season. We thus apply the perturbation only throughout January 2000 and explore its impact over the subsequent years. We have chosen the period 2000-2008 because it is marked by a resumption in the formation and storage of relatively large

(a)

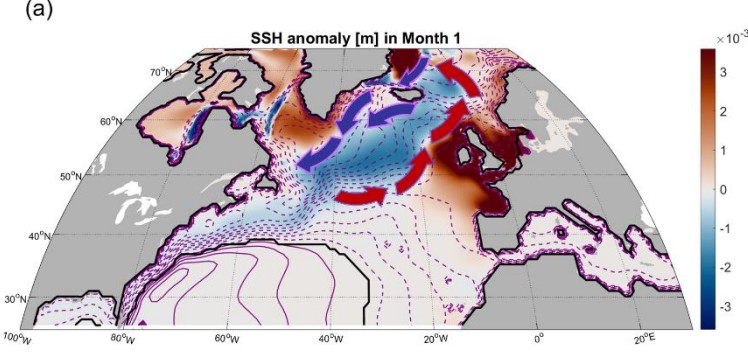

b)

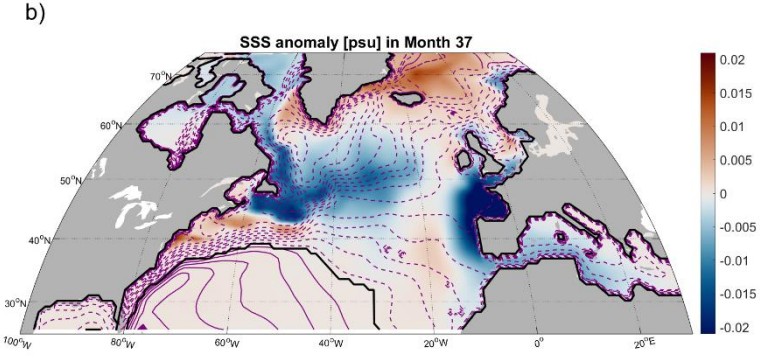

c)

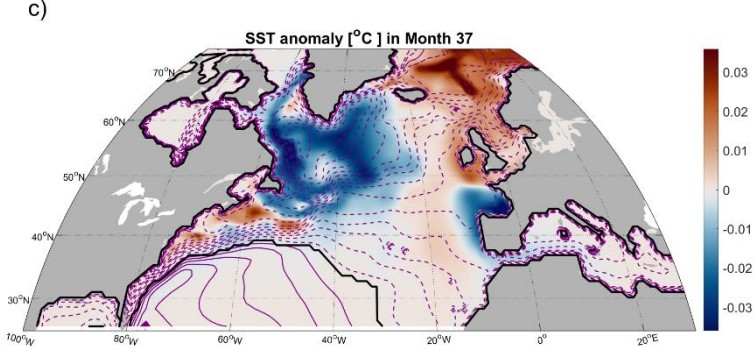

**Figure 6: (a) Monthly-mean SSH anomaly ([m] shaded, positive in red, negative in blue) from Month 1 of the freshwater perturbation experiment relative to the unperturbed ECCO state estimate. The freshwater perturbation is sustained throughout Month 1. Superimposed are the corresponding SSH contours from the unperturbed ECCO state estimate, 7 cm apart. Positive contours are solid purple lines, negative contours are dashed, and the zero contour is a thick black line. Schematic arrows indicate the impact of the SSH anomalies on the anomalous circulation pattern: increased northeastward transport towards the Iceland basin and increased southward transport along the East Greenland Shelf; (b) Same as in a but shading indicates the contemporaneous sea-surface salinity (SSS) anomaly [psu] in Month 37 of the experiment (36 months after the freshwater perturbation) and the corresponding SSH contours from the unperturbed ECCO state estimate. (c) Same as in b but shading indicates the SST anomaly [°C].**

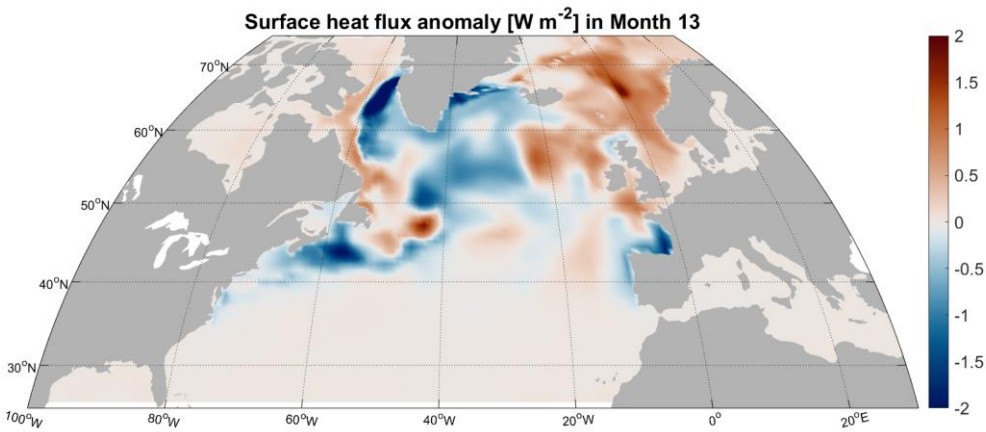

**Figure 7: Monthly-averaged surface heat flux anomaly $[W\ m^{-2}]$ in Month 13 of the experiment, twelve months after the surface freshwater perturbation applied in January 2000 of the ECCOv4r2 state estimate. Blue shading indicates anomalous surface heat flux from the atmosphere to the ocean, and red shading corresponds to anomalous transfer of heat from the ocean to the atmosphere.**

volumes of LSW (Fig. 2). Hence, launching our experiment in 2000 allows us to explore this regime of enhanced LSW production. We adjust the amplitude of our positive and negative values in the applied perturbation pattern such that the net input of freshwater is zero. Adjusting the poles of the applied forcing pattern is more physically consistent than redistributing the freshwater imbalance as a uniform area-averaged offset over the North Atlantic. Unlike a uniform redistribution of the imbalance, the poles of the sensitivity pattern are aligned with dynamical barriers such as inter-gyre boundaries.

We use our freshwater flux perturbation experiment to explore the adjustment of the subpolar North Atlantic and the response of LSW volume on intermonthly and interannual timescales (e.g., one, three, and five years after the surface perturbation). When comparing against the unperturbed ECCO state estimate, we see that the "Traffic Controller" pattern affects the SSH gradients in the subpolar gyre (Fig. 6a). In the climatology (contour lines in Fig. 6a), the SSH decreases in the direction towards the Labrador Sea and Southeast Greenland. In contrast, the SSH anomaly in our experiment is more positive in the Labrador Sea and more negative along the NAC pathway (Fig. 6a). Contours of the SSH anomalies can be used to infer changes in the geostrophic component of surface currents (Jones et al., 2023). It is also important to note that in this freshwater flux experiment, we keep wind stress unperturbed, and hence do not change ageostrophic wind-driven Ekman transport relative to the control state. Therefore, the SSH anomalies in Fig. 6a tell us that our "Traffic Controller" perturbation *decelerates* NAC transport towards the western subpolar gyre (blue arrow in Fig. 6a) relative to the control run. However, the "Traffic Controller" increases northeastward transport towards the Iceland Basin and the GIN Seas (red arrow in Fig. 6a) compared to the control simulation. In addition, there is an increase in the southward transport along the East Greenland Shelf (Fig. 6a).

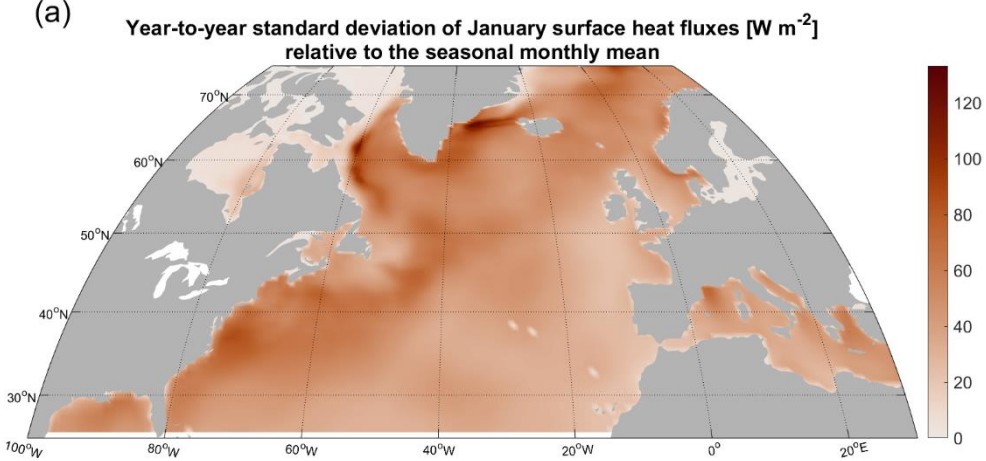

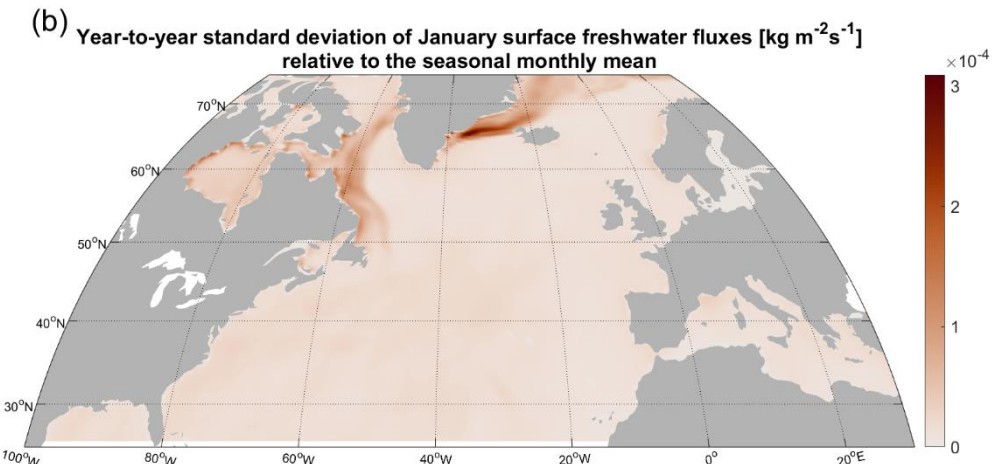

**Figure 8: (a) Year-to-year standard deviation of the January surface heat fluxes $[Wm^{-2}]$ in the ECCOv4r4 state estimate ; (b) same as in a but for surface freshwater fluxes $[kg\ m^{-2}\ s^{-1}]$**

Our results also show that sea-surface temperature (SST) throughout the subpolar gyre decreases in response to the perturbation, while sea surface salinity (SSS) exhibits a dipole pattern: freshening along the western boundary of the Labrador

Sea and salinification along the Greenland shelf (Fig. 6b,c). Early on in the experiment, the circulation anomaly in the subpolar gyre causes more intense surface heat loss in the Iceland Basin and the Nordic Seas (Fig. 7). This surface heat loss anomaly is a signature indicating that relatively more warm water of subtropical origin has penetrated the Nordic Seas compared to the

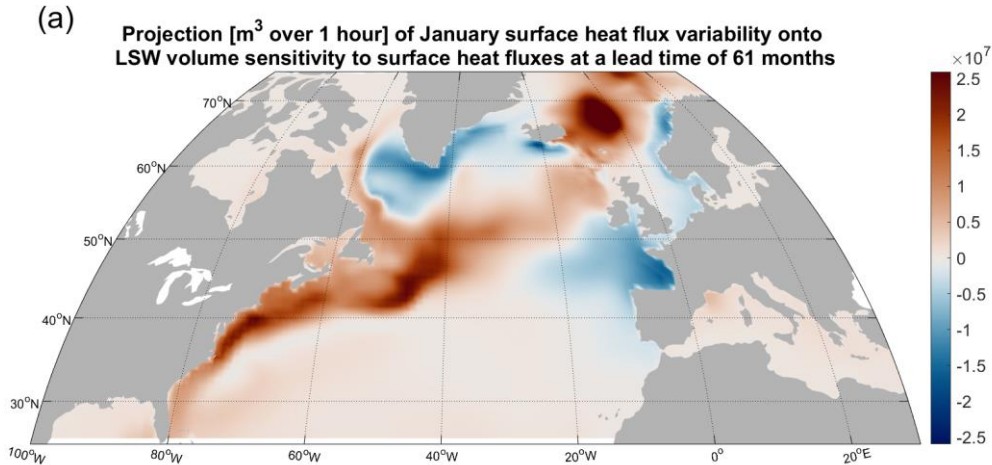

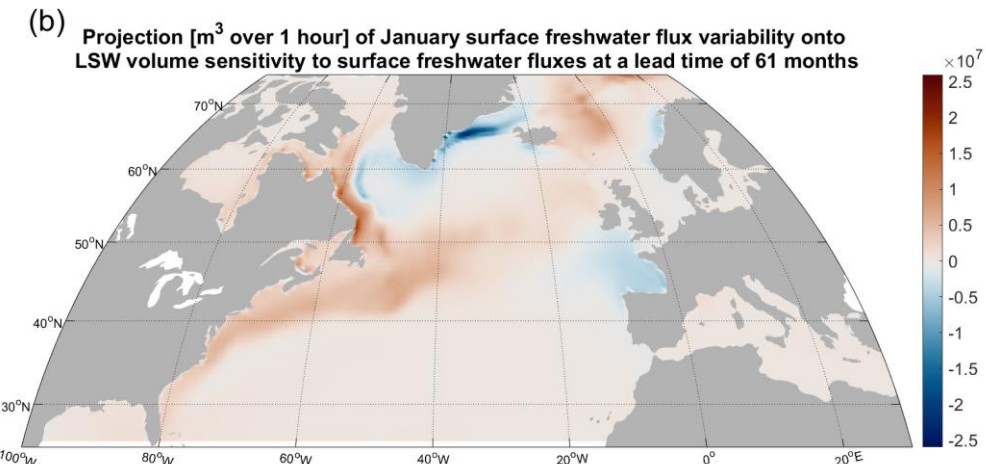

**Figure 9: (a) Projection [$m^3$ over 1 hour] of the surface heat flux variability from Fig. 8a onto the sensitivity of LSW to surface heat fluxes at a lead time of 61 months from Fig. 4g. (b) Projection [$m^3$ over 1 hour] of the surface freshwater flux variability from Fig. 8b onto the sensitivity of LSW to surface freshwater fluxes at a lead time of 61 months from Fig. 4h.**

control simulation. Anomalous heat loss is also an additional feedback mechanism promoting densification of the surface water. Three years (36 months) after the perturbation, we see the southward transport of anomalously denser, relatively warmer but more saline water (Fig. 6b,c) from the GIN Seas to the Irminger Sea through the Denmark Strait. Southwest of Denmark Strait, we see anomalously colder and more saline water at the surface three years after the applied perturbation.

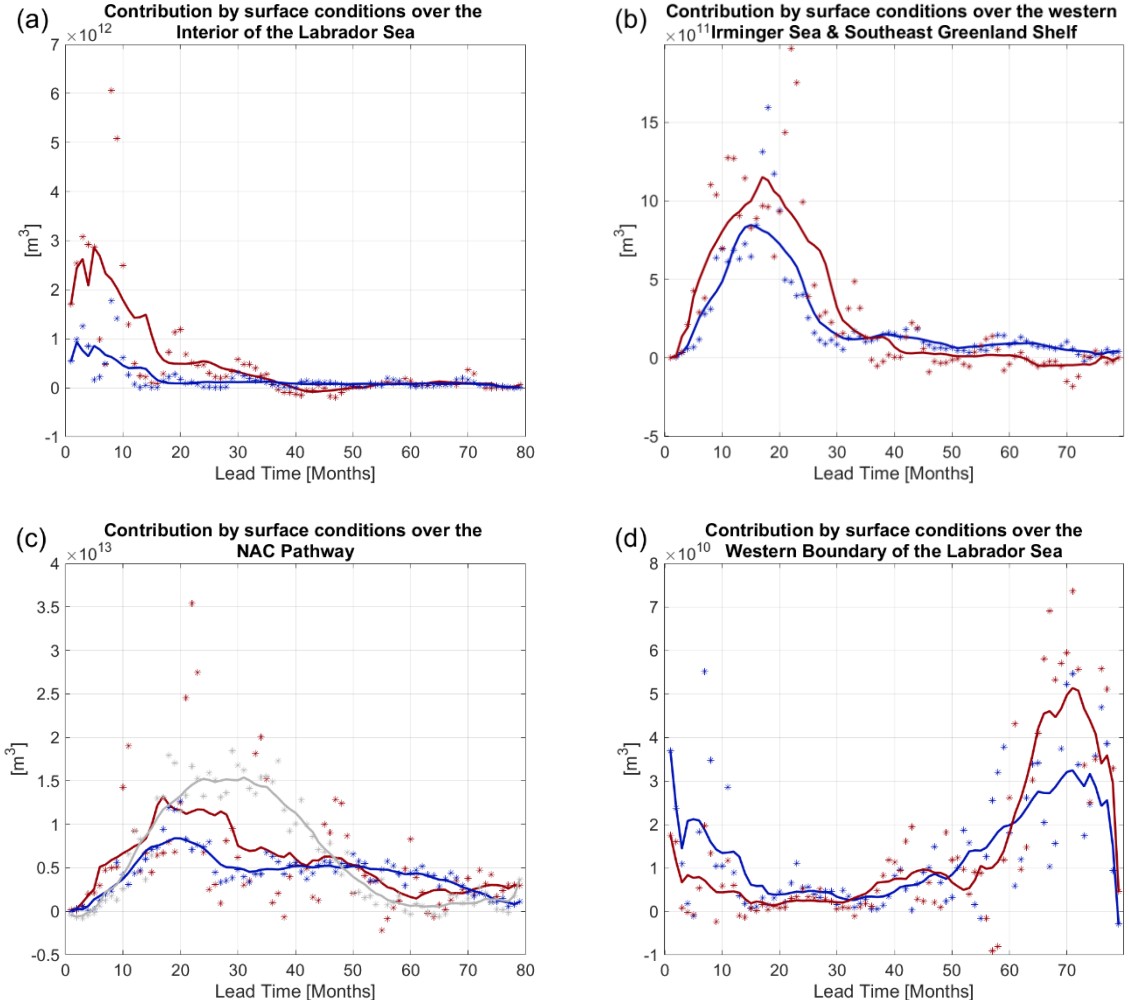

Figure 10: Preferred lead times in the contributions of surface boundary conditions over the regions shown in Fig. F1 to LSW variability $[m^3]$ in ECCOv4r4, Eq. (11): (a) the Labrador Sea interior (the region with bottom depth larger than 2.5 km); (b) the western part of the Irminger Sea and the southeast Greenland shelf; (c) the NAC pathway over the region defined in Appendix F and Fig. F1; and (d) Labrador Sea Western Boundary, defined as the area shallower than 2 km and south of 59°N. The horizontal axis denotes the lead time. Shown are the regional contributions of surface heat fluxes (red), freshwater fluxes (blue), and zonal wind stress (gray, panel c only) to intermonthly variability in LSW volume. Asterisks correspond to monthly data points, while the solid lines represent running means averaging the data points over 13-month windows. Only sensitivity patterns from spring 2006 – winter 2007 objective functions are used in the calculation.

The initial adjustment of the subpolar gyre leads to more intensive production of LSW in the Irminger Sea and south of Cape Farewell (Fig. E1a, corresponding to April, 15 months after the perturbation). Some of this water with LSW properties is advected towards the Labrador Sea along the path of the Deep Western Boundary Current (DWBC) and increases the volume of stored LSW even in the late summer (Figs. E1a and E2). As a result of the applied freshwater flux pattern, the East Greenland Current (EGC), the Irminger Current (IC), and the West Greenland Current (WGC) transport anomalously cold and saline

water from the Irminger to the Labrador Sea (Fig. 6 and schematic in Fig. 1). Finally, there is enhanced production and storage of LSW in the Labrador Sea itself (Fig. E1b,c). As a result, the volume of accumulated LSW in the Labrador Sea increases (Figs.

E1b,c and E2) on interannual timescales. Using Eq. (7) and surface boundary conditions from the surface freshwater flux experiment, we are largely able to recover the evolution of the LSW volume anomaly in the basin (Fig. E2). This serves as additional independent verification (Fig. E2) of our reconstruction methodology. Both the experimental output and our reconstruction (Figs. E1 and E2) show that the "Traffic Controller" pattern of surface freshwater fluxes impacts LSW on a timescale of several years.

However, the "Traffic Controller" pattern is an abstract construct: a sensitivity to hypothetical surface forcing anomalies. To assess whether this genuinely gives rise to LSW volume variability, we also need to know whether there is actual historical variability in the ECCO surface boundary conditions that projects onto this sensitivity pattern and activates it. To that end, we consider the standard deviation of surface heat and freshwater fluxes to obtain a map of surface variability for each month of the year relative to the seasonal cycle (Fig. 8). As expected for a high latitude region, the variability is larger in

the winter season compared to the summer (not shown). The pattern of surface flux variability is also intensified near the western boundaries of the basins. In particular, the winter-time surface freshwater fluxes exhibit some of the strongest variability along the western boundary of the Labrador Sea (Fig. 8b) in the marginal sea-ice zone (Petit et al., 2023). We project the wintertime surface buoyancy flux variability patterns onto the corresponding "Traffic Controller" sensitivity patterns to see where heat and salt fluxes contribute the most to the delayed response of LSW volume with a lag of 5 years (Fig. 9). Once

again, the western boundaries of the North Atlantic and its marginal seas appear to be a prominent source of LSW variability at a lead time of 5 years (Fig. 9b).

We also explore if there are preferred lead-lag relationships in the system by looking at weighted lagged correlations (Eq. (11)) where anomalies in the surface boundary conditions lead in time, and reconstructed LSW volume exhibits a lagged response. As expected, we find that the contributions of surface heat and freshwater fluxes over the Labrador Sea interior

(bottom depth larger than 2.5 km), over the Southeast Greenland Shelf, and over the Western Irminger Sea exhibit characteristic lead times that peak within two years (Fig. 10a,b and Fig. F2a,b). In comparison, surface fluxes of heat, freshwater, and momentum along the NAC pathway contribute to LSW volume anomalies both at lead times shorter than a year but also with a lag greater than 2 years (Fig. 10c and Fig. F2c). The large magnitude of LSW volume variability generated by boundary conditions along the NAC further highlights the importance of this advective pathway, especially at interannual lead times

(Fig. 10c and Fig. F2c). Surface buoyancy fluxes along the Labrador Sea Western Boundary (sea floor shallower than 2 km) contribute to an even more delayed responses, peaking at lead times greater than 4 years (Fig. 10d and Fig. F2d). This is consistent with the findings of Kostov et al. (2022) who suggest that LSW volume responds to surface perturbations along the Labrador Sea Western Boundary on timescales set by the propagation of signals from the western to the eastern subpolar gyre and back.

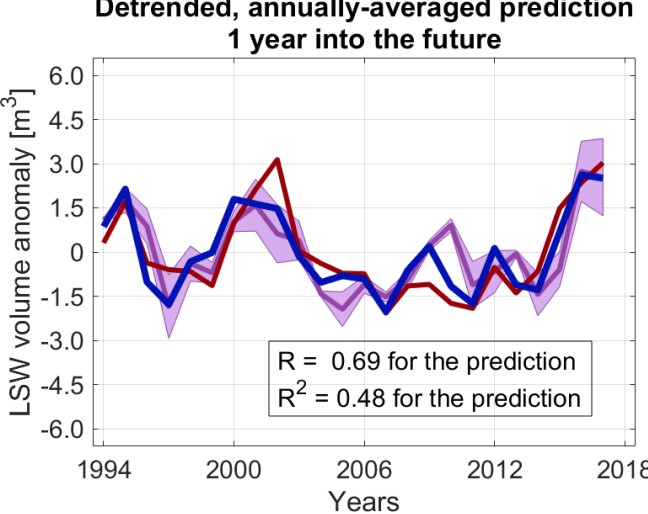

**Figure 11: Annually-averaged prediction (dark purple) of LSW volume anomalies relative to the annual mean at a time horizon of 1 year into the future. The shaded envelope reflects the uncertainty of the prediction estimated analogously to the uncertainty in Fig. 3f. Superimposed is the full reconstruction from Fig. 3f (blue) that also includes information about surface boundary conditions at zero lag. The actual ECCOv4r4 anomalies in LSW volume relative to the annual cycle are shown in red. The reconstruction skill (Pearson correlation coefficient R=+0.69) for a prediction one year into the future, shown here, is much larger than the reconstruction skill for a prediction two years into the future (not shown).**

Surface boundary conditions over different regions make contributions of very different magnitudes to intermonthly (Fig. 10) and interannual (Fig. F2) LSW variability. Among the four regions that we focus on, surface boundary conditions along the NAC pathway make the largest contribution (Figs. 10c and F2c). This further highlights the importance of the "Traffic Controller" pattern that we identify and its role in driving LSW volume anomalies via alterations to the strength and the pathway of NAC transport.

The existence of such long characteristic lead times in the system motivates us to explore the predictability of LSW variability into the future. We analyze the predictability of annually-averaged LSW volume anomalies at a time horizon of 1 year by applying the modified convolution from Eq. (8). We thus omit information about the surface boundary conditions at zero lag, and when reconstructing LSW anomalies in each year, we only include past information from the preceding years. We show that using historical information alone, we can predict 48% of LSW volume variability in ECCOv4r4 (Pearson correlation coefficient R=0.69) a year into the future (Fig. 11). This skill is not sufficient for making reliable long-range predictions of watermass anomalies but demonstrates the propensity of LSW for a delayed response to surface forcing away from the interior of the Labrador Sea. Our skill decreases drastically if we extend our annual-mean prediction two years into the future (not shown). This sharp decline in prediction skill is not a surprise, as the responses to surface heat and momentum fluxes along the NAC exhibit peaks at lead times of roughly 1.5 to 2.5 years (Fig. 10c and F2c).

## 4. Discussion and Conclusion

In this study, we have presented a novel linear reconstruction of accumulated LSW volume in the Labrador Sea using *only* surface boundary conditions: wind-stress and surface heat and freshwater fluxes. In addition, we offer a *causal* attribution of historical LSW variability. Our results suggest that wind stress, freshwater fluxes, and heat fluxes make contributions of comparable magnitude to LSW volume anomalies (Fig. 3a,d). These novel results challenge the traditional view that wintertime cooling in the Labrador Sea is the dominant driver of interannual variability in LSW volume in the Labrador Sea.

We furthermore show that surface boundary condition anomalies away from the Labrador Sea can trigger statistically significant delayed responses of LSW. For example, the watermass volume is sensitive to meridional wind stress along the eastern boundary of the North Atlantic, all the way to Western Africa. A wave signal from this region is quickly communicated along the coastal waveguide to the subpolar gyre (Pillar et al. 2016; Jones et al., 2018; Kostov et al., 2021; Loose et al., 2020). Our findings are thus similar to the results of Jones et al. (2018), who highlight similar enhanced sensitivity of total heat content in the Labrador Sea to wind stress along the African and West European shelves.

Our results also indicate that the LSW volume is particularly sensitive to remote zonal wind and buoyancy forcing along the pathway of the NAC but also on its flanks, extending westwards towards the Labrador Sea and eastwards towards the European Basin. We refer to the geographical fingerprint of this remote sensitivity as the "Traffic Controller" pattern: in effect, SSH and surface density anomalies aligned with the NAC branches can accelerate or decelerate the currents and divert them away from or towards the Irminger Sea, the Iceland Basin and the Nordic Seas. Similarly, the input of momentum by surface winds also impacts the NAC branches. This modulates the transport of warm, saline subtropical water that loses buoyancy in the Iceland Basin and the Irminger Sea (Petit et al., 2020; Petit et al., 2021) and is subsequently transformed into LSW.

Some general circulation models show a significant lagged correlation between the upper AMOC limb and LSW, where the former leads the latter in time (Ortega et al., 2017; Li et al., 2019). Although this lagged relationship has been highlighted in previous literature, we are the first to identify it as a causal chain and an oceanic teleconnection in a state estimate constrained with observations (Forget et al., 2015). Our "Traffic Controller" sensitivity pattern is a geographical fingerprint associated with this oceanic teleconnection that relates flow along the upper AMOC limb to LSW formation and storage in the Labrador Sea. Surface wind stress and density anomalies can act to divert and redirect the transport of warm and saline subtropical water which is necessary for the formation of LSW in the Labrador and Irminger Seas.

The "Traffic Controller" sensitivity pattern appears to be activated on a regular basis by large SSS variability originating along the Western Boundary of the Labrador Sea. This same region has also been shown to be an important driver of subtropical AMOC variability (Kostov et al., 2021; Kostov et al., 2022). The Labrador Sea Western Boundary is a marginal sea ice zone, where changes in freezing, melting, and the ice distribution strongly affect the salinity budget of the upper ocean. The model representation of these ice-driven SSS fluctuations is sensitive to the resolution of the ocean domain and may be a significant source of bias in historical simulations (Petit et al., 2023).

We show that the contribution of SSS variability in the Labrador Sea marginal sea ice zone to LSW volume anomalies is a delayed response with a preferred lead-time of more than 3 years. This is consistent with the experiments in Kostov et al. (2022) who suggest that freshening along the western boundary of the basin impacts LSW with a lag of five years.

Notably, surface momentum and buoyancy fluxes along the NAC pathway also trigger anomalies in LSW volume with lags on a timescale of 2 to 5 years. The existence of such large delayed responses in the ocean system implies that surface conditions along the NAC pathway are a major source of LSW variability.

We have tested the predictability of LSW anomalies a year into the future using only past information about surface boundary conditions. We demonstrate that we can estimate approximately 48% of LSW variability at a time horizon of 1 year into the future. This limited but substantial predictive skill at a 1-year horizon suggests that local conditions in the Labrador Sea at zero lead time are not the only important driver of LSW volume anomalies. Our results imply that preconditioning in the North Atlantic away from the Labrador Sea contributes to driving watermass variability. In particular, our results suggest that anomalies in the NAC transport of heat and salt modulate surface-forced watermass transformation in the eastern subpolar gyre (and the GIN Seas), which then affects the production of LSW downstream in the Irminger and Labrador Seas. This agrees with the findings of Petit et al. (2021) who show that air-sea fluxes in the eastern subpolar gyre dominate the variability of low-stratified mode water that acts as a precursor to the formation of LSW. We therefore suggest that our reconstructions and predictions should not be judged merely on the basis of their skill. Our analysis sheds light on the importance of various physical processes that affect the accumulation of LSW in the Labrador Sea.

The scope of our study is limited by the methods we use. We rely on linear response theory which may not hold in the case of regime shifts in the North Atlantic circulation. More generally, we do not account for nonlinearity in the LSW response. In addition, we compute our objective function, LSW watermass volume, only over particular historical periods from the ECCO state estimate. We expect that linearizing the model about an earlier or later period may produce different sensitivities of LSW to surface boundary conditions. For example, the LSW formation sites and export pathways may change in time. This impact of the background ocean state on the sensitivity of LSW is reflected in the envelope of uncertainty in our reconstructions. Similarly, including a longer time history of past surface forcing can affect our results. Other limitations come from the fact that we use a general circulation model to estimate the sensitivity of LSW to boundary conditions. Our lagged sensitivity patterns inevitably depend on the model dynamics and any bias therein. For example, the model configuration in this study does not resolve eddies, which can significantly impact ocean transport and variability in the subpolar gyre (e.g., Zhao et al., 2018).

Such potential sources of model bias are mitigated by the fact that we use the ECCO configuration of the MITgcm, a state-estimate which is constrained by available historical observations in a least squared sense (Forget et al, 2015). This gives us confidence that our analysis is relevant to real-world ocean dynamics and gives an insight into processes affecting the volume budget of the important LSW watermass. In this study we focus on the watermass volume budget, but the same approach can be extended to a future analysis of LSW heat content variability.

## Appendix A. Activation Functions

In our definition of LSW volume as a numerical objective function (Eq. (5)), we employ logistic activation functions to approximate Boolean conditions. For example, the condition that LSW potential density $\sigma_\theta > \sigma_{\theta\,lower} = 27.7\text{kg m}^{-3}$ (Eq. (1)) is expressed as

$$(1/2 + \tanh\left((\sigma_\theta - \sigma_{\theta\,lower})\,c\right)/2),\ c = 10^5\ kg^{-1}\ m^3 \tag{A1}$$

The above expression asymptotes to zero as $\sigma_\theta$ decreases below the lower bound $\sigma_{\theta\,lower} = 27.7\text{kg m}^{-3}$ and asymptotes to 1
as $\sigma_\theta$ increases above $\sigma_{\theta\,lower}$ (see Fig. A1a for an illustrative example of an activation function). Because the constant factor $c = 10^5\ kg^{-1}\ m^3$ is very large, the activation function (Eq. (A1)) has a very steep slope near $\sigma_{\theta\,lower}$. An offline calculation shows that the activation functions in Eq. (5) give the same result for the volume of LSW in the Labrador Sea in ECCO as the Boolean conditions (Eq. (1) and (3)) within numerical precision.

The derivative of our activation functions with respect to the input is approximately zero everywhere except in a range
of input values close to the imposed upper and lower bounds that define LSW:

$\sigma_{\theta\,lower} = 27.7\text{kg m}^{-3},\ \sigma_{\theta\,upper} = 27.84\text{kg m}^{-3},\ \text{PV}_{upper} = 4 \times 10^{-12}\ m^{-1}s^{-1}$

(see for example the shaded area in Fig. A1). Therefore, our objective function, as defined in Eq. (5) has nonzero derivatives only when a model grid cell in the Labrador Sea reaches $\sigma_\theta$ or $PV$ values near these thresholds. In these transitional ranges, the activation functions have a maximum slope of $c/2$, equal to $5 \times 10^4\ kg^{-1}\ m^3$ or $5 \times 10^4\ m\ s$ in our case.

The steep maximum slope of the activation function raises the question whether the chosen large values of $c = 10^5\ kg^{-1}\ m^3$ and $c = 10^5\ m\ s$ affect the derivatives of the objective function. Have we arbitrarily rescaled the sensitivity patterns by factor $c$? In this context, an important point to consider is that in our adjoint calculations, the objective function is not evaluated during a single model timestep of 1 hour, but is averaged over a time period corresponding to 1 simulated month. Hence, Labrador Sea grid cells that enter or leave the LSW potential density and potential vorticity range spend many model
timesteps gradually evolving across the transitional regime near the $\sigma_{\theta\,lower}$, $\sigma_{\theta\,upper}$, and/or $\text{PV}_{upper}$ thresholds (for example, the shaded region in Fig. A1). In addition, the objective function is defined over a large spatial region, the entire Labrador Sea. When a model grid cell in the region of interest enters or leaves the LSW potential density or potential vorticity range, this transition involves adjacent grid cells, too. The objective function that we define averages temporally over 1 month and spatially over each group of grid cells that transition from one watermass to another.


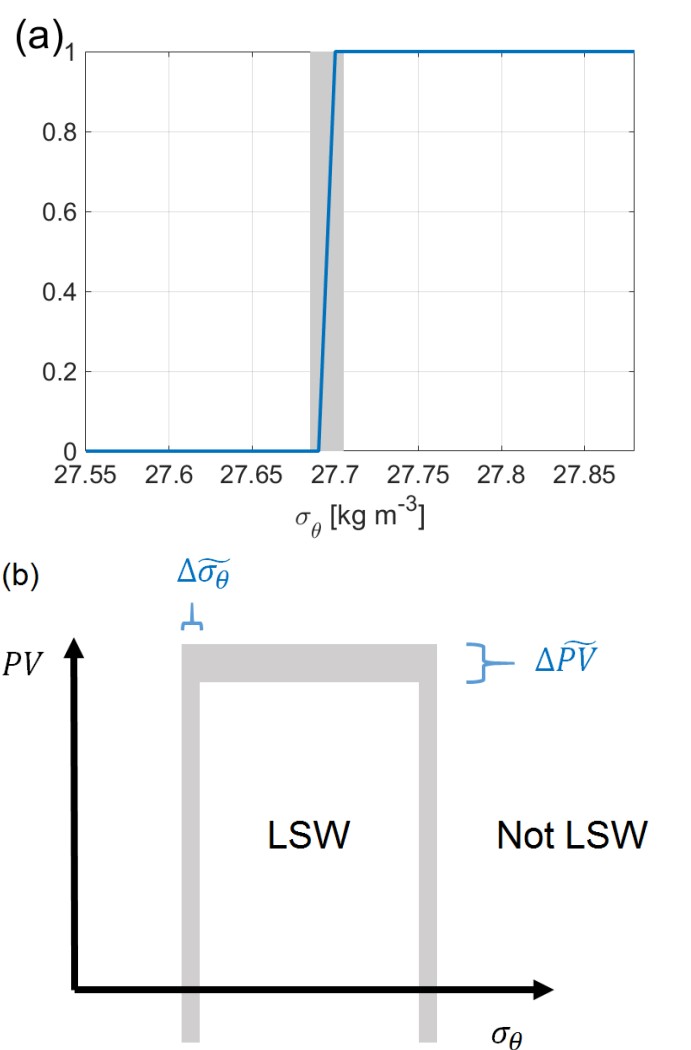

Figure A1: (a) An illustrative example of a logistic activation function that imposes the condition $\sigma_\theta > \sigma_{\theta\,lower} = 27.7\ kg\ m^{-3}$. Gray shading indicates a transition region where the value of the activation function is between 0 and 1. Over this region, the average slope of the activation function is $\approx 1/$(width of the shaded area) in units of $kg^{-1}\ m^3$ ; (b) a schematic of the definition of LSW in potential density and PV space with the transition regions shown with gray shading as in (a). The activation functions have a slope close to 0 outside the transition regions.

Modifying factor $c$ changes only the weight that we assign to a grid cell undergoing transformation relative to its neighboring grid cells and the weight of some days of the month in which we average the objective function relative to other days of the same month. However, the average weight of all transitioning cells remains the same even if we change factor $c$. Given enough temporal and spatial data points throughout the transition region (e.g., shaded area in Fig. A1), the slope of the applied activation function always averages to 1/(the width of a transitional range) across the transitional regime between LSW and non-LSW water. That is because the logistic activation function always increases from $\approx 0$ to $\approx 1$ in the transition range,

as shown in Fig. A1 (shaded area). This result holds irrespective of the value of parameter $c$ and the maximum slope $c/2$, so

long as the slope of the activation function is close to zero outside the transition range. Thus, we do not expect that in the case of our monthly-averaged basin-wide objective functions, the order of magnitude of $c$ will affect the order of magnitude of the sensitivity patterns.

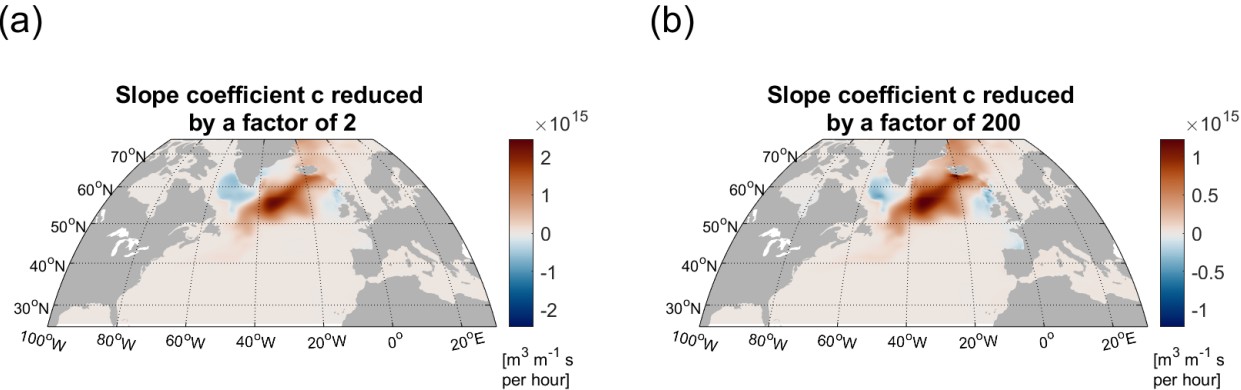

**Figure A2: (a) Sensitivity of the LSW volume to surface freshwater flux anomalies $[m^3 m^{-1} s$ over 1 hour] at a lead time of 2 years, where the objective function is defined using the default activation functions with $c = 5 \times 10^4 \ kg^{-1} \ m^3$ for potential density and with $c = 5 \times 10^4 \ m \ s$ for potential vorticity, respectively. (b) same as in a but with activation function parameters set to $c = 5 \times 10^2 \ kg^{-1} \ m^3$ for potential density and with $c = 5 \times 10^2 \ m \ s$ for potential vorticity. The limits of the colorbar in a are exactly twice as large as the colorbar limits in b.**

We test this assumption by conducting a set of adjoint calculations with a value of $c$ varied by different orders of magnitude between $c = 5 \times 10^4 \ kg^{-1} \ m^3$ and $c = 5 \times 10^4 \ m \ s$ on one hand, and $c = 5 \times 10^2 \ kg^{-1} \ m^3$ and $c = 5 \times 10^2$ $m \ s$ on the other hand. We otherwise keep all model settings the same. We compare the results for the lagged sensitivity of LSW volume to surface freshwater flux with the new results with reduced $c$ (Fig. A2). We see that over a range of lead times, the sensitivity patterns with different choices of $c$ remain almost identical in their magnitude and geographical fingerprint. The

difference between the corresponding patterns is much smaller – roughly a factor of two – which is orders of magnitude smaller than the factor of 100 by which we vary $c$ (Fig. A2). This dependence on the choice of factor $c$ is also smaller than the change in the adjoint sensitivity patterns and adjoint-based reconstructions that we see when we linearize over different time periods from the ECCO historical simulation (e.g., the different individual reconstructions in Fig. 3c). This gives us confidence that the arbitrary choice of $c$ does not affect the validity of our results regarding the volume of LSW, but at the same time we

caution that our approach is not universally valid for any objective function of interest. In other words, there are ocean indices for which our method may not be applicable. In our case, changing the value of $c$, may reduce or increase the skill of our reconstruction, but we have not explored the possibility of tuning this parameter. Instead, we have focused on explaining the physical mechanisms revealed by our analysis.

**Appendix B. Atmospheric and oceanic regimes in the ECCO state estimate**

To compute the lagged sensitivity of LSW volume to surface boundary conditions, we linearize the model trajectory about particular background states of the system. When we compute wintertime LSW volume as our objective function, we consider the years 2006, 2007, and 2011 in ECCOv4r2. These years represent three different states of the North Atlantic Oscillation (NAO, Fig. B1) based on the National Oceanic and Atmospheric Administration (NOAA) NAO index (https://www.cpc.ncep.noaa.gov/products/precip/CWlink/pna/nao.shtml). The NAO is an atmospheric regime that influences

subpolar variability (Dickson et al. 1996; Rhein et al., 2017; Roussenov et al., 2022). Similarly, these three years correspond to different winter mixed layer depths in the Labrador Sea and different volumes of LSW. Within our small ensemble, the skill of our linear reconstructions seems to be strongly related to the background LSW volume in the year when we compute the objective function. The timeseries showing the most extreme fluctuations in Fig. 3c corresponds to a reconstruction that uses the sensitivity of wintertime LSW volume in 2006 to past surface boundary conditions.


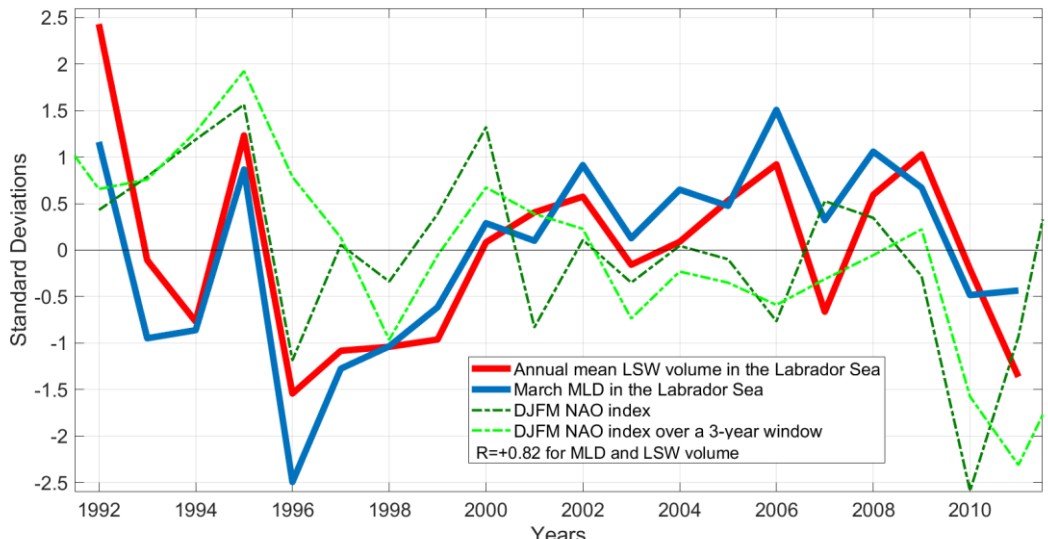

**Figure B1: Annual mean LSW volume in the Labrador Sea (red) in ECCOv4r2; March mixed layer depth in the Labrador Sea (blue) in ECCOv4r2; December-January-February-March NAO index (dark green) from the NOAA database (https://www.cpc.ncep.noaa.gov/products/precip/CWlink/pna/nao.shtml). A positive NAO phase is associated with lower sea level**

**pressure above Iceland; December-January-February-March NAO index from the NOAA database (dark green) averaged over each year and the preceding 2 years (light green). All indices were normalized by the standard deviation, and the time-mean was subtracted.**

Our small ensemble size does not allow us to comment on the ability of the NAO to influence our reconstruction skill. However, we see a strong correlation between winter NAO variability and variability in the Labrador Sea March mixed layer

depth, as well as annual-mean LSW volume anomalies (Fig. B1).

## Appendix C. Assumed memory of past surface boundary conditions

In our reconstructions, we have to assume a cut-off lead time $t_{cutoff}$ on a timescale of years even though the ocean retains memory of past surface boundary conditions on much longer timescales. We are limited by computational resources and by

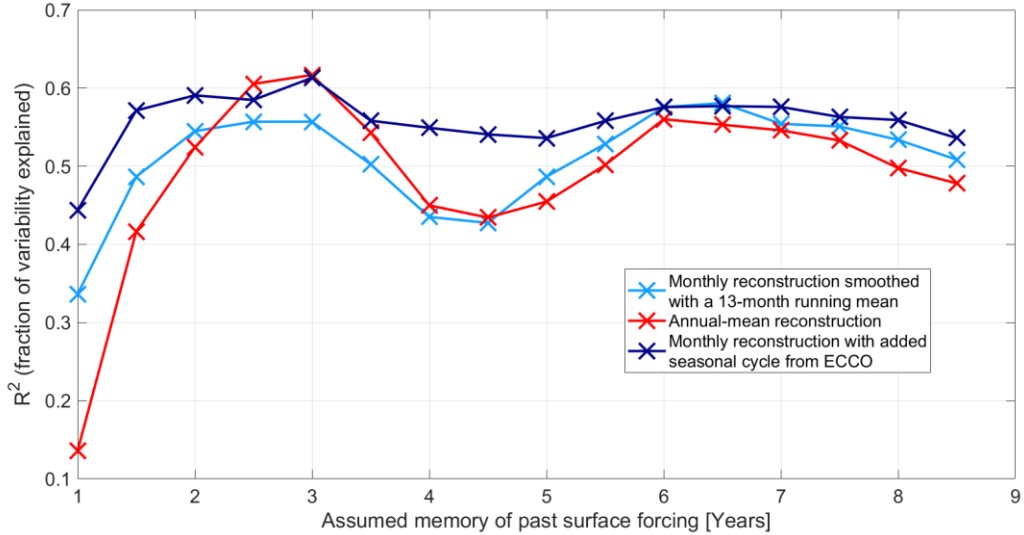

**Figure C1: Sensitivity of the reconstruction skill with respect to the assumed memory of past surface forcing (cut-off lead time $t_{cutoff}$). The reconstructions here use only adjoint sensitivities from objective functions representing LSW volume over the spring 2006 – winter 2007 time period. Light blue crosses show the skill of monthly reconstructions smoothed with a 13-month running mean that recover anomalies relative to the seasonal cycle of ECCOv4r2. Red crosses correspond to annual-mean reconstructions relative to the long-term mean. Dark blue crosses show the skill of monthly reconstructions superimposed on the time-mean seasonal cycle from ECCOv4r2, compared against the monthly timeseries from the state estimate.**

the fact that the ECCOv4r2 run, which we linearize about, starts in 1992. Last but not least, the adjoint linearization at high latitudes may become less reliable on long timescales.

We test how our reconstruction skill changes as we vary the cut-off lead time $t_{cutoff}$ (Fig. C1). We find that both the monthly and annual-mean reconstructions improve noticeably if we take into account surface boundary conditions going more than 2.5 years back in time. The reconstruction skill declines if we assume a memory of surface conditions longer than 6.5 years (Fig. C1). We thus set the cut-off lead time $t_{cutoff} = 6.5$ years for all points in the timeseries where data for the preceding 6.5 years are available from ECCO.

## Appendix D. Sensitivity of LSW to boundary conditions at lead times shorter than a year

At lead times shorter than a year, LSW volume is particularly sensitive to local surface buoyancy anomalies in the Labrador Sea itself (Fig. D1c,d). The sensitivity to zonal winds is also most pronounced in the Labraror Sea itself (Fig. D1a), but is marked by spatial noise.

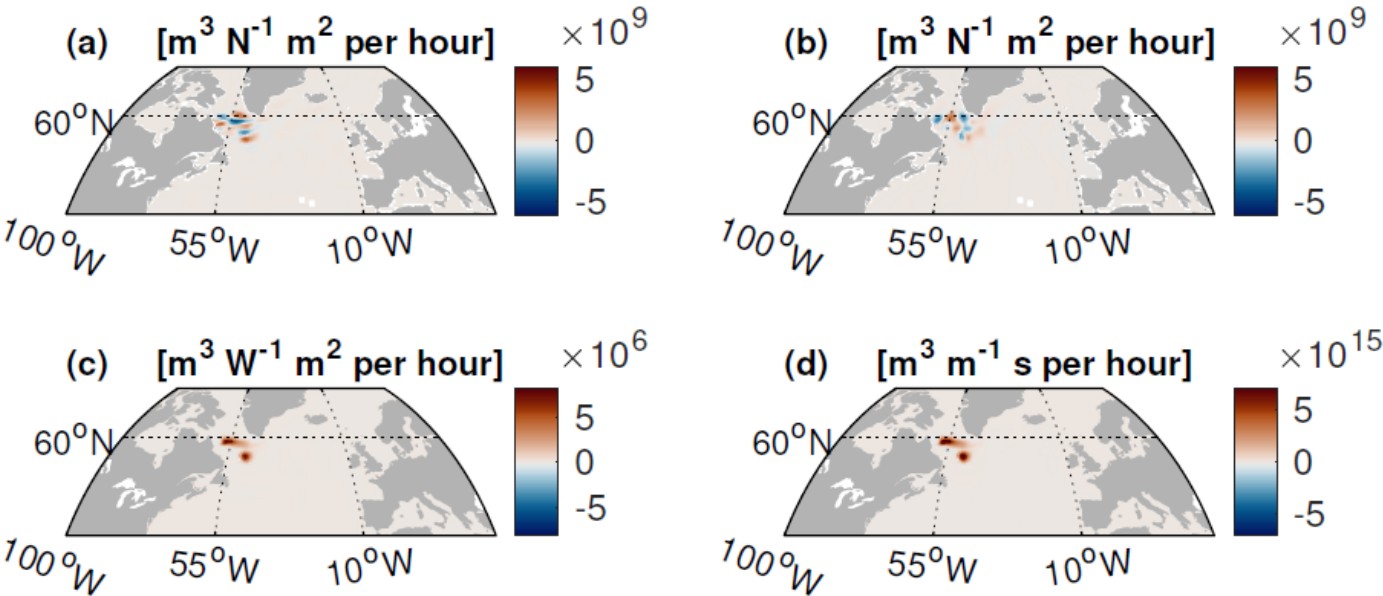

**Figure D1: Sensitivity of wintertime LSW volume in 2007 to surface boundary conditions at a lead time of 1 month. Sensitivity to zonal (positive eastward) (a) and meridional (positive northward) (b) windstress $[m^3 N^{-1} m^2$ over 1 hour], surface heat fluxes (c) $[m^3 W^{-1} m^2$ over 1 hour] out of the ocean, and surface freshwater fluxes $[m^3 m^{-1} s$ over 1 hour] out of the ocean (d). Red shading indicates that a positive anomaly in the surface boundary condition leads to an increase in LSW volume at a lead time of 1 month.**

## Appendix E. Adjustment of the North Atlantic Subpolar Gyre in response to the imposed surface freshening and salinification perturbation

We explore in greater detail the ocean's response to the freshening and salinification perturbation applied in January 2000 of the ECCO state estimate and discussed in Section 3. The adjustment of the subpolar gyre gives rise to a dipole in surface heat fluxes with anomalous cooling over the Iceland Basin a year into the experiment (Fig. 7). This preconditions enhanced LSW formation. We first see an increase in LSW production in the Irminger Sea (Fig. E1a), then a thickening of the LSW layer near Cape Farewell (Fig. E1b), and finally, a thickening of the LSW layer across the Labrador Sea (Fig. E1c,d). After an initial transient decline in LSW volume, these processes eventually give rise to an overall increase in the volume of LSW stored in the Labrador Sea on a timescale of several years after the perturbation (Fig. E2). Notice that the LSW volume anomaly in the Labrador Sea (Fig. E2) reaches a peak during summer months, 2.5 years after the applied perturbation, and some of the LSW thickening in the Labrador Sea is very pronounced near the southeast Greenland Shelf (Fig. E1b). This is consistent with the explanation that a fraction of

the additional LSW is not produced locally in the Labrador Sea but is imported as an anomaly from the Irminger Basin following

an advective pathway around southern Greenland. At the same time, there are indications that there is a reduction in the southward

and eastward export of LSW (blue shading in Fig. E1b) relative to the unperturbed state. Furthermore, the cold anomaly in the

Labrador Sea (Fig. 6c) likely acts to decrease the rate of seasonal restratification of LSW.

a)

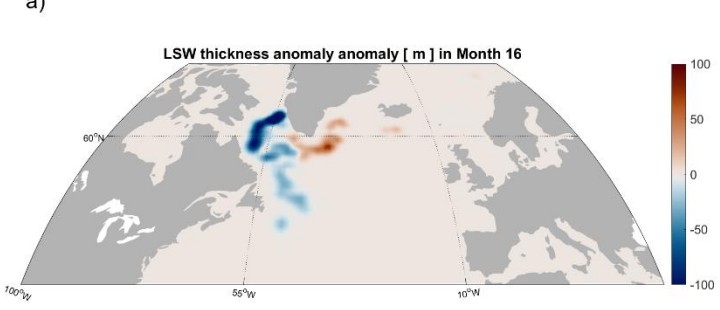

b)

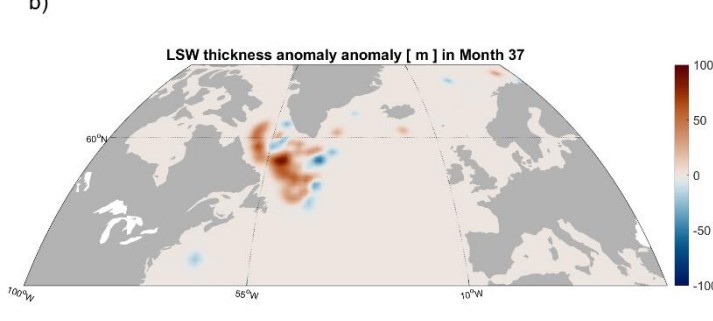

c)

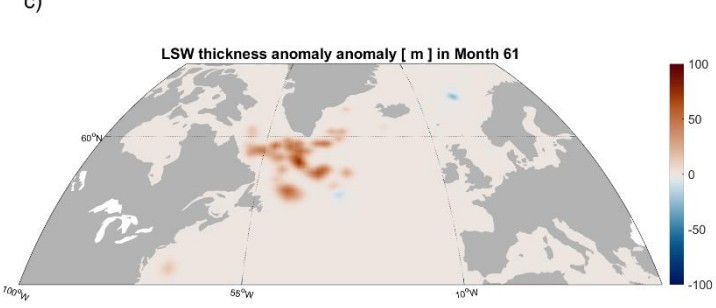

775

**Figure E1: Time-evolving anomaly in the thickness [m] of the LSW layer in response to the freshwater flux perturbation pattern applied in January 2000 of the ECCO state estimate. The panels correspond to monthly averages at Months (a) 16; (b) 37; and (c) 61 of the freshwater perturbation experiment, where the perturbation is applied throughout Month 1.**

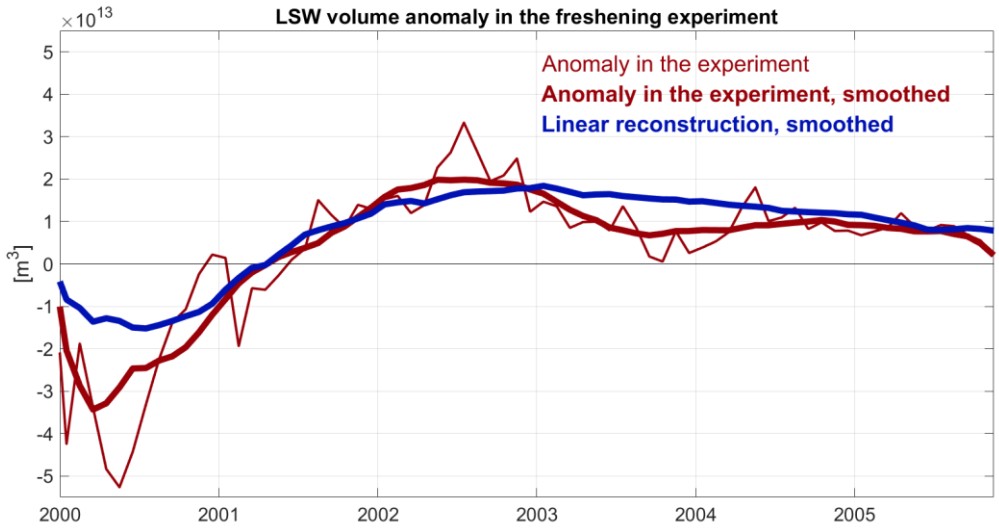

**Figure E2: Time-evolving LSW volume anomaly [$m^3$] in the Labrador Sea in response to the freshwater flux perturbation pattern applied in January 2000 of the ECCO state estimate (thin red line) ; Same but smoothed with a 13-month running mean (thick red line) ; Linear reconstruction of the LSW volume anomaly in the perturbation experiment (thick blue line), also smoothed with a 13-month running mean.**

## Appendix F. Definition of regions analyzed in the study

We define the interior of the Labrador Sea as the part of the basin where the bottom depth reaches below 2.5 km (dark blue shading in Fig. F1). We also select a region north of 60ºN and west of 30ºW, encompassing the western part of the Irminger Sea and the southeast Greenland Shelf as illustrated in Fig. F1.

We furthermore define an NAC pathway region as the area bounded by the -0.75 cm and -0.35 cm climatological SSH contours between 50°W and 30°W, and between 40°N and 58°N in the North Atlantic (purple shading in Fig. F1). The NAC flows northeastward across this region.

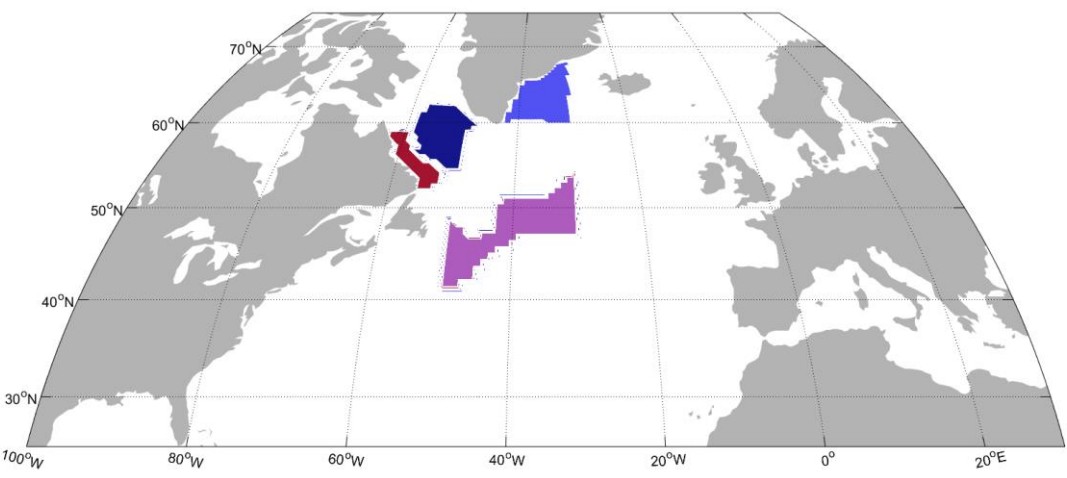

 **Figure F1: Spatial masks of the regions used in identifying characteristic lead times: the Labrador Sea interior (dark blue); the western Irminger Sea and the southeast Greenland Shelf (light blue); the NAC pathway region (purple); the Western Boundary of the Labrador Sea (red). The masks are set to 1 in a given region of interest and 0 everywhere else.**

Finally, we define a Labrador Sea Western Boundary region as the basin area with a depth shallower than 2 km and located south of 59°N (red shading in Fig. F1).

In Fig. 10, we consider the preferred lead times at which the regions in Fig. F1 contribute to intermonthly variability in LSW volume. In comparison, in Fig. F2, we consider the preferred lead times in the contributions to interannual LSW variability.

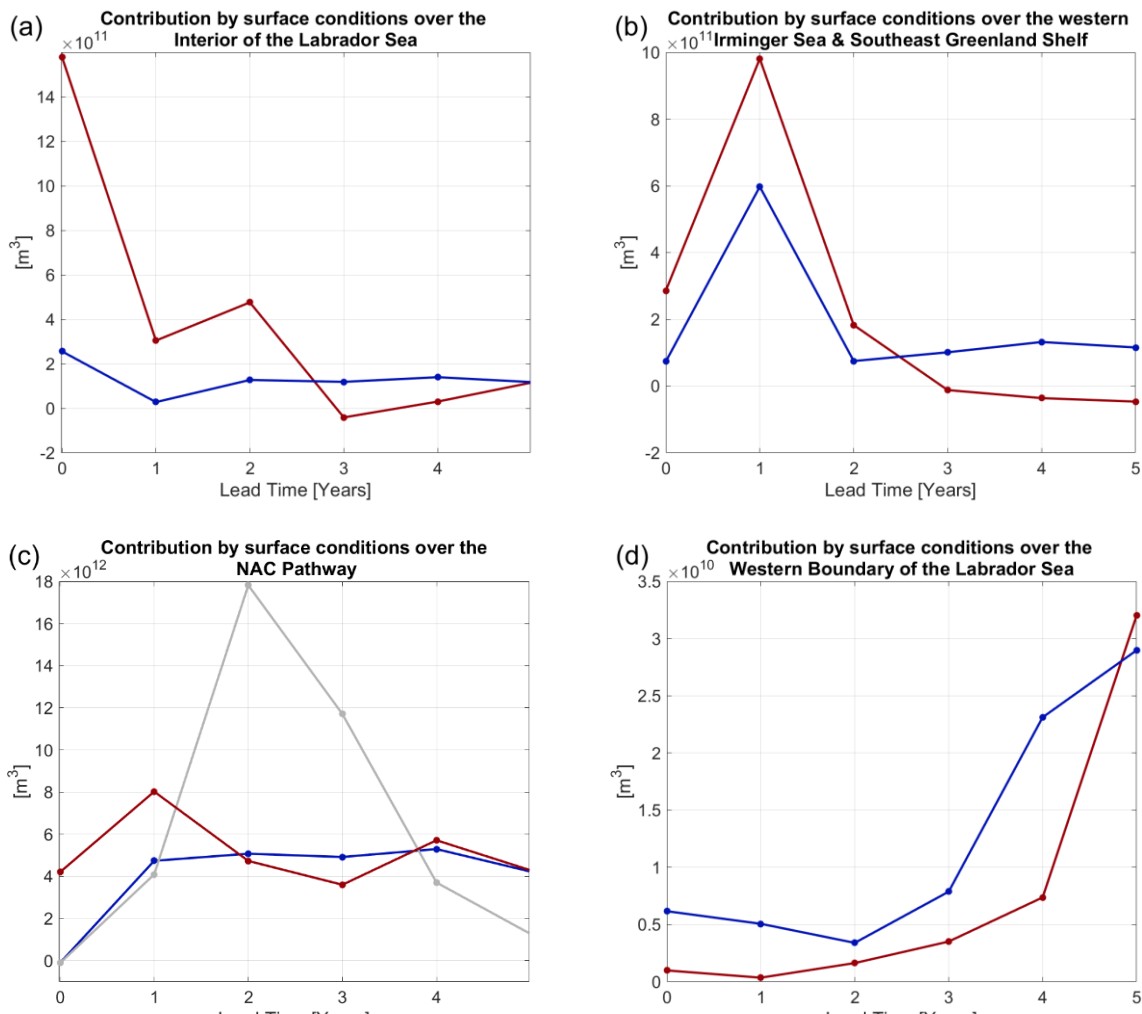

**Figure F2: Same as Fig. 10 but for regional contributions of surface boundary conditions over the regions shown in Fig. F1 to interannual variability in ECCOv4r4. Preferred lead times in the regional contributions of surface boundary conditions of LSW variability $[m^3]$, Eq. (11) over (a) the Labrador Sea interior (the region with bottom depth larger than 2.5 km); (b) the western part of the Irminger Sea and the southeast Greenland shelf; (c) the NAC pathway over the region defined in Appendix F and Fig. F1; and (d) Labrador Sea Western Boundary, defined as the area shallower than 2 km and south of 59°N. The horizontal axis denotes the lead time. Shown are the regional contributions of surface heat fluxes (red), freshwater fluxes (blue), and zonal wind stress (gray, panel b only) to interannual variability in LSW volume. Only sensitivity patterns from spring 2006 – winter 2007 objective functions are used in the calculation.**

**Acknowledgements**

We thank the ECCO, OSNAP, SNAP-DRAGON, and TICTOC research groups for the helpful discussions. The numerical simulations used in this study were conducted on the ARCHER2 supercomputer and on Exeter University's ISCA supercomputer. We express our appreciation to the group that perpetually maintains the ECCO state estimate. We also thank Ralf Giering from FastOpt for the algorithmic differentiation software TAF. The color map vik (Crameri 2018) is applied in this study to provide accessibility to readers with color-vision deficiencies (Crameri et al., 2020). YK and MJM were funded

by NERC TICTOC grant NE/P019064/1 and NERC BLT grant NE/S001433/1. HM was funded by the French National Center for Scientific Research (CNRS). DPM was supported by SNAP-DRAGON grant NE/T013494/1 and TICTOC grant NE/P019218/1. HLJ acknowledges SNAP-DRAGON grant NE/T013494/1 and WISHBONE grant (NE/T013451/1).

**Declarations**

The authors declare no competing interests.

**Author contributions**

All authors contributed to the conception, development, design, and writing of this study.

**Data availability**

The MITgcm code is available at https://github.com/MITgcm/. The ECCO state estimate model configuration is accessible at https://github.com/gaelforget/ECCOv4. ECCO's initial and boundary conditions are made available

at https://web.corral.tacc.utexas.edu/OceanProjects/ECCO/ECCOv4. TAF is a proprietary algorithmic differentiation software provided by FastOpt. The repository https://github.com/YavorKostov/LSW-Volume/ provides access to Fortran and Matlab code used in this study, as well as binary input files for the model calculations. All reasonable requests for relevant information should be addressed to the corresponding author.

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
