# Peer review of "Surface factors controlling the volume of accumulated Labrador Sea Water"

_EGUsphere, 2023_

## Author Comment (AC1)

Referee #1

We thank Referee #1 for the balanced and thorough review that touches upon multiple aspects of our study and helps us improve our manuscript.

All line numbers in this response refer to the updated manuscript.

This manuscript aims to explore historical variability of Labrador Sea Water volume and the processes that drive that variability. The authors use the ECCO state estimate and its adjoint, in combination with an objective function to reconstruct and attribute variability. Although in the discussion the authors indicate the study has multiple limitations based around the assumption their approach makes, the authors show that the volume of LSW accumulated in the Labrador Sea exhibits a delayed response to surface wind stress and buoyancy forcing outside the convective interior of the Labrador Sea (e.g. the NAC). They use this response to predict a fraction of the LSW variability a year in advance. They also suggest that all of wind stress, freshwater fluxes, and heat fluxes make contributions of comparable magnitude to LSW volume anomalies, rather than wintertime cooling being the dominant driver.

Given the importance of the Labrador Sea and its deep water formation, looking at drivers of its variability is important. The new results, if robust, will help us understand past changes in the Labrador Sea, as well as its future evolution. The paper is also generally well written, with good quality figures. Thus, the work is eventually worth publishing.

Thank you for pointing out the positives and the strengths of the manuscript!

Before then, I do have some concerns about the authors approaches and assumption that need to be addressed before it can be published. I also think the background literature needs to be improved and the work better sited in what we already know about the Labrador Sea.

We have expanded the literature review in the introduction (lines 32-35, 41-59, 69-77, 124-130, 135-140).

Thus. I would recommend major revisions.

The introduction feels short and underwhelming for such a full length manuscript. There is less than a page of general material before the authors begin to delve into their own approach and plans for the manuscript.

We have expanded the literature review in the introduction (lines 32-35, 41-59, 69-77, 124-130, 135-140). The introduction is now more than twice as long as it was before.

I would like to see more background on the Labrador Sea and LSW formation. What do we know about the processes that drive its formation and variability to help put the author's work in context. Especially important is a well-developed literature over the past ~20 years looking at processes that drive LSW formation, from idealized models, to water mass transformation approaches, to numerical models.

We have expanded the literature review on the topic of LSW formation in the newly added lines 41-59

Many of these studies have looked at the role of winds,

We have discussed the role of winds in lines 55-59 of the introduction.

the buoyancy forcing (including separating the two components – the heat and freshwater forcing) – yet I don't see a single reference to any such studies.

We refer to previous literature on the role of surface cooling and freshwater forcing in the newly added lines 41-52 in the introduction.

And although the authors focus is on air-sea forcing, lateral exchange from the boundary currents to the interior are also important in balancing the air-sea forcing, yet I don't see any discussion of these processes and their potential relevance.

We have added references and discussed lateral exchange in the newly added lines 57-59 and 72-74.

Also, later on the authors bring in the role of forcing in regions like the EGC, the NAC, but yet there really isn't a discussion of the sub-polar gyre circulation and how all these pieces are linked.

Later on, in lines 478-485 we have now emphasized the role of particular currents for transporting temperature, salinity, and density anomalies. We have also pointed more explicitly to our schematic diagram of ocean currents (now separated as a standalone Fig. 1).

To truly understand the significance of the authors work and what is now, a reader needs that missing background.

We have expanded the background review in the introduction (lines 32-35, 41-59, 69-77, 124-130, 135-140).

The authors use the ECCO state estimate for their analysis. I feel the authors assume the reader is very familiar with this product. Which is likely not always the case. More background is needed, especially discussion of the quality of its representation of the SPG and the Labrador Sea, so give the reader confidence in its being the authors underlying tool.

We have expanded our description of ECCO in lines 136-140 and 150-162, including details about the framework (lines 150-158) and the quality of its representation of the SPG and Labrador Sea (lines 158-162).

Also, what years were ECCO run over? I understand it limits how far back the authors can go. But why does their study stop in 2012. The latest version of ECCO goes to 2017 I believe. And given the strong convection in the Labrador Sea (and its shift to the east) in 2012, and 2015-2018, it seems to be a major loss to ignore those recent years.

In the first submission of this manuscript, we used ECCOv4 release 2, which ended in December 2011. Following your comment, we have redone our calculations using the subsequent ECCOv4 release 3 and release 4. In the updated manuscript, we show results from ECCOv4 release 4

which gives us a timeseries of LSW variability from 1992 until the end of 2017 (e.g., Fig. 2, Fig. 3).

Another important question is the authors definition of LSW. How was the range 27.7-27.84 arrived at? It doesn't seem to fit any common definition I've seen in the literature. Many studies break it down into upper LSW and classical LSW, but those ranges are typically 27.68-27.74 and 27.74-27.8. The various Yashayaev papers argue there are various 'vintages' of LSW and one can't use a fixed density range through time (Feucher et al show how much difference that can make in a model that include salinity drift). Other works (especially models) define a higher upper bound because of those drift (i.e. a model LSW). Do the authors pick their LSW range based on ECCO's behavior? If so, that is fine, but the choices must be explained and justified, with discussion of why the range differs from other studies. I'd also like to see some sensitivity analysis related to that range. And then discussion of how such a range may impact the results (in the discussion section).

As we now say in lines 124-127, "Our main constraint for defining LSW is based on vertical stratification (a component of the potential vorticity, PV), while we also define generous potential density bounds to help identify the watermass. This is similar to the definition of LSW used in Li et al. (2019).."

We have cited Feucher et al. in line 185 (and our updated manuscript also cites 5 papers where Yashayaev is a first author and several co-authored papers on LSW).

Most importantly, we have elaborated on the choice of density range in lines 170-192, where we state:

"Following Talley and McCartney (1982), Zou and Lozier (2016) and Li et al. (2019), we approximate PV in terms of the vertical stratification:

$$PV \approx f \frac{N^2}{g}$$
(2)

where $f$ is the Coriolis parameter, $g$ is the gravitational acceleration, and $N$ is the buoyancy frequency. Thus, the PV condition which we impose ensures that we define LSW as a weakly stratified watermass. Li et al. (2019) demonstrate that in observations and in most models, this criterion is universal and sufficient for identifying LSW in the Labrador Sea. However, in some models and model configurations, low stratified water can be found below the LSW layer in the basin (Li et al., 2019). Thus, we introduce a second constraint, which sets bounds on the potential density $\sigma_\theta$ of the watermass referenced to the surface:

$$\sigma_{\theta\, lower} < \sigma_\theta < \sigma_{\theta\, upper}$$
(3)

where $\sigma_{\theta\, lower} = 27.7 \text{kg m}^{-3}$ and $\sigma_{\theta\, upper} = 27.84 \text{ kg m}^{-3}$. This potential density constraint is deliberately very generous because in ECCO, similarly to many models and observations, the density, temperature, and salinity of the LSW formed in the Labrador Sea differ from year to year (Feucher et al., 2019). However, we have tested a stricter density constraint with $\sigma_{\theta\, upper} = 27.81 \text{ kg m}^{-3}$. Using $27.81 \text{ kg m}^{-3}$ as the upper bound gives the same results for the volume of LSW in release 2 of ECCOv4. In contrast, in the subsequent release 3, the Labrador Sea in ECCO has a secondary deep layer of low-stratified water denser than $27.81 \text{ kg m}^{-3}$ and

distinct from the LSW above. The existence of this deep low-stratified water explains most of the time-mean offset between our calculation of LSW volume in ECCOv4 release 3 compared to releases 2 and 4. In the most recent release 4, there is only a brief period between the mid to late 1990s where the model produces and then stores low-stratified water all the way down to 27.84 kg m$^{-3}$ in the Labrador Sea. Outside this period both $\sigma_{\theta\ upper} = 27.84$ kg m$^{-3}$ and $\sigma_{\theta\ upper} = 27.81$ kg m$^{-3}$ give the same results for the volume of LSW in ECCOv4 release 4."

As described in the new text above, we have done a sensitivity analysis with respect to this density range. This sensitivity analysis would take up too much space if we were to include it in the manuscript or the appendices, but we are providing it here for your reference. The figures below illustrate that in ECCOv4 release 2, $\sigma_{\theta\ upper} = 27.84$ kg m$^{-3}$ and $\sigma_{\theta\ upper} = 27.81$ kg m$^{-3}$ give almost identical results. In release 4, using $\sigma_{\theta\ upper} = 27.84$ kg m$^{-3}$ or $\sigma_{\theta\ upper} = 27.81$ kg m$^{-3}$ gives the same result except in the late 1990s. The reason behind this good agreement is that our main criterion for defining LSW is actually PV rather than density. In effect, LSW is defined as water with low vertical stratification. This definition has been promoted by Zou and Lozier (2016) and Li et al. (2019), cited in the manuscript.

In contrast to the other releases, in ECCOv4 release 3, there is a secondary deep layer of low-stratified water, which would correspond to NEADW rather than LSW. This creates a larger discrepancy between using $\sigma_{\theta\ upper} = 27.84$ kg m$^{-3}$ or $\sigma_{\theta\ upper} = 27.81$ kg m$^{-3}$. This issue, however, is not present in release 2 or release 4.

[Figure]

Additional Figure: Sensitivity analysis of the LSW definition with respect to density, while using the same PV criterion.

I also wonder about the choices of the time period of the winter objective function and the two winter functions. Firstly, today, is it really not possible to broadly compute the functions for all years and seasons. Yes, it would take longer but I would have thought the authors could have gotten some supercomputer access to do so. But if it is not possible to go beyond these 3 periods, I am still wondering about their choice. 2006, 2007 and 2011 are all towards the end of the ECCO timeseries used.

We had limited computational resources early on and currently no access to a computational budget for this project. We have added two more runs with summertime objective functions (e.g., the new Fig.2, Fig. 3, Figure 11), but we cannot extend the ensemble further by considering objective functions in additional years.

In lines 233-237, we have now justified the choice of years towards the end of the ECCOv4 release 2 timeseries for defining objective functions: "The selected three years are close to the end of the ECCOv4 release 2 state estimate because that allows us to compute lagged sensitivity over longer periods leading up to these three years. We use the sensitivity patterns for the ECCOv4 release 2 objective functions to reconstruct not only LSW variability in release 2, but also to attempt reconstructing LSW in the more recent ECCO release 3 (not shown) and in release 4, that extends further in time until the end of 2017."

And although ECCO has significant LSW formation in 2006, I don't believe that year is shown to have high LSW in the observations, such as some of the Yashayaev papers. March 2008 would have been a better strong LSW formation winter.

Unfortunately, March 2008 would not have been a very representative month in ECCOv4. Lines 373-375 now say: "However, all releases of ECCOv4 seem to underestimate the magnitude of the 2008 relative increase in LSW, where the underestimation is most pronounced in release 4."

My next major questions resolves around the forcing functions that the authors look at – wind, heat and freshwater. It looks like the authors are comparing each term to the LSW formation. But for example, sensible heat fluxes depend significantly on the wind. How is that taken into account and attributed? At the very least, this needs to be discussed to help the reader understand what the authors mean by wind or heat forcing, for example.

Following your recommendation, we have first clarified this in lines 119-123:

"We treat the surface boundary conditions in such a way as to avoid overlap and double-counting between their interrelated contributions. For instance, when we analyze the impact of surface winds, we account for their input of momentum. In contrast, the winds' impact on air-sea heat exchange is considered to be part of the heat flux contribution to LSW variability. Similarly, when we talk about the effect of surface heat fluxes, we do not include their impact on surface salinity via evaporation, as that is accounted for in the contribution of surface freshwater fluxes."

We have also expanded the paragraphs in lines 272-279 that discusses this. We now clarify

"Therefore, both the net surface heat flux $Q$ and the default sensitivity patterns $S(Q, x, t − t_{lead})$ output by the adjoint include air-sea feedback. As a result, the convolution in Eq. (7) can erroneously double count air-sea feedback mechanisms.

In order to avoid this problem, we cannot rely on the default configuration of the adjoint. Instead, we have to instruct the algorithmic differentiation software not to take derivatives of the bulk formulae and the surface radiation parameterization code. This approach guarantees that our lagged sensitivity patterns do not include air-sea feedback effects that are already accounted for in the net surface heat flux budget and the net surface freshwater budget. For example, the effect of surface heat fluxes on surface salinity via evaporation is accounted for only in the surface freshwater budget. In addition, the impact of evaporation and precipitation on surface temperature via latent heat fluxes is accounted for only in the surface heat flux budget. However, following our approach, the model's forward trajectory remains the same as in the optimized ECCO state estimate."

Beyond this, the authors show a breakdown based on these 3 forcing mechanisms but there is little text and discussion. Figure 2 has lots of interesting signals that get a superficial discussion at best.

Figure 2 is now Figure 3, and we have added further discussion in lines 371-378 following your recommendation: "For example, salt fluxes, heat fluxes, and wind stress, all, contribute to the positive LSW volume anomalies in the early 2000s. We suggest that the 2008 relative increase in LSW explored by Yashayaev and Loder (2009) can be attributed primarily to surface heat fluxes (Fig. 3a). These attribution results hold true across our three reconstructions. However, all releases of ECCOv4 seem to underestimate the magnitude of the 2008 relative increase in LSW, where the underestimation is most pronounced in release 4. We find that the subsequent 2010-2011 decline in LSW volume is dominated by heat fluxes and wind stress, while the 2012 recovery is attributable to both heat and salt fluxes (Fig. 3a). The well-documented increase in LSW volume after 2015 (Yashayaev and Loder, 2017) seems to be primarily related to wind stress anomalies and, to a smaller extent, surface heat fluxes."

Also, the heat and freshwater spatial flux plots are shown using different units – maybe convert them both to buoyancy flux components, so the reader can more easily compare the magnitude and significance of the terms.

Rescaling the sensitivity to heat and freshwater fluxes in terms of buoyancy fluxes would not be as trivial as it may seem, and here is a brief explanation of the challenges. The rescaling of linear sensitivity patterns would involve thermal expansion and haline contraction coefficients. The issue is determining robustly which local and seasonal coefficients are appropriate to use. The spatial patterns represent seasonal sensitivity of LSW, which occupies intermediate depths in the Labrador Sea. Yet, the fluxes to be rescaled enter the surface over a wide area that extends all the way to the recirculation gyre and the subtropical gyre and in a different season depending on the lead time. If we want to rescale the fluxes into units of buoyancy, we need to use some values for the local coefficients of thermal expansion and haline contraction appropriate for a particular season. It is not so trivial to determine whether the linearized coefficients have to be representative of the deep Labrador Sea in the season of the objective function. Another possibility is that the coefficients have to be representative of the locations along the surface where the fluxes penetrate the ocean in the season corresponding to the lead time. A third option is some weighted average of the equation of state coefficients spanning the pathway between the surface penetration region and the deep Labrador Sea and averaging over a range of seasons.

The case in the present manuscript is particularly challenging because the objective function is evaluated in a region of intensive water mass transformation. This is a major difference

compared to the Kostov et al. (2019) study, where sensitivities of the *subtropical* AMOC to surface fluxes near Greenland were qualitatively rescaled into units of buoyancy. If the reviewer expresses further interest, the first author can explain in further detail why the Kostov et al. (2019) rescaling into buoyancy units was not as challenging.

Adding such a calculation in the present manuscript would require making more assumptions and introducing additional technical explanations about the approximations. The text is already quite loaded with technical explanations because of the nature of the adjoint calculations. We decided against introducing this further complication.

Why rescale to a perturbation order of 10^-8? Why not to order 1, to make the numbers simpler?

We have now clarified this in lines 437-439: "However, we multiply the pattern by a factor of $(-10^{-22})$, so that the rescaled perturbation is of order $10^{-8} \, m \, s^{-1}$ (or $\sim 10^{-5} \, kg \, m^{-2} \, s^{-1}$), which is comparable to the standard deviation in January surface freshwater fluxes between different years (see Fig. 8b)."

The authors define their sensitivity pattern as a "Traffic Controller". I assume they are hoping this will make the concept easier for the readers to understand. But honestly, I had trouble seeing that acronym and got confused during that discussion. More discussion and a focus to make the definition clear for all readers is needed.

We have now tried to introduce this "nickname" earlier in the text. In lines 17-19, in the abstract, we say: "stress and buoyancy forcing outside the convective interior of the Labrador Sea, at important locations in the North Atlantic Ocean. In particular, patterns of wind and surface density anomalies can act as a "traffic controller" and regulate the North Atlantic Current's (NAC) transport of warm and saline subtropical water masses that are precursors for the formation of LSW."

In addition to all other instances where the "Traffic Controller" is mentioned, we have also added lines 538-540: "This further highlights the importance of the "Traffic Controller" pattern that we identify and its role in driving LSW volume anomalies via alterations to the strength and the pathway of NAC transport."
Following your suggestion, we also refer more to the "Traffic Controller" in the Discussion (lines 579-585): "Some general circulation models show a significant lagged correlation between the upper AMOC limb and LSW, where the former leads the latter in time (Ortega et al., 2017; Li et al., 2019). We are the first to identify this as a causal relationship and an oceanic teleconnection in a state estimate constrained with observations (Forget et al., 2015). Our "Traffic Controller" sensitivity pattern is a geographical fingerprint associated with this oceanic teleconnection that relates flow along the upper AMOC limb to LSW formation and storage in the Labrador Sea. Surface wind stress and density anomalies can act to divert and redirect the transport of warm and saline subtropical water which is necessary for the formation of LSW in the Labrador and Irminger Seas."

Also in the applied perturbation experiments, over exactly what region were the patterns applied?

We have now specified the perturbed region in lines 436-437: "an extended North Atlantic region (north of 20°N, west of 20°E, south of Fram Strait, including marginal seas but excluding the Mediterranean and the Baltic)."

Given the salinification along the Greenland shelf, is there a link to Fram Strait and Arctic outflow/processes?

We have added text in lines 478-480 that describes the importance of temperature and salinity anomalies advected through Denmark Strait:

"Three years after the perturbation, we see the southward transport of anomalously denser (warmer but more saline, Fig. 6b,c) water from the GIN Seas to the Irminger Sea through Denmark Strait."

However, on the short timescales that we explore, Arctic processes and transport changes through Fram Strait do not play a role.

I also feel there is too much material in the Appendices – too many times the reader is referred to a figure in the appendices, which has relevant material for understanding the main text. For example, the definitions of the regions. I think such information could be included on other figures.

The main text is already busy with figures that convey a lot of information. As a result, we have not moved more graphical information from the Appendices to the main text. However, we have taken note of your concern about the figure panels defining the various regions in Fig. F1 in Appendix F. Following your advice and that of Reviewer 2, we have merged the panels that define regions of interest into one single-panel Fig. F1 where individual regions are distinguished by color. We have also reduced the number of panels in Fig. E1 in Appendix E, where the emphasis now is on the response 36 months after the perturbation.

---

## Author Comment (AC2)

Referee #2

We thank Referee #2 for the detailed and exhaustive review of our study and for the valuable feedback.

All line numbers in this response refer to the updated manuscript.

Review of "Surface Factors Controlling the Volume of Accumulated Labrador Sea Water" by Kostov, Messias, Mercier, Marshall, and Johnson

The manuscript reconstructs accumulation and variability of the Labrador Sea Water (LSW) using the ECCO state estimate. Specifically, it examines contributions of local and remote surface momentum and buoyancy fluxes to the LSW variability. In general, the present findings are very similar to those reported in several previous studies.

We have expanded the literature review in the introduction (lines 32-35, 41-59, 69-77, 124-130, 135-140) citing previous studies that have used observations and/or models of various degrees of complexity to explore the climatology and variability of LSW.

As we acknowledge in lines 131-136: "A number of previous studies have applied the adjoint of the MITgcm to exploring sources of ocean variability (e.g., Czeschel et al., 2010; Heimbach et al, 2011; Pillar et al., 2016; Jones et al., 2018; Smith and Heimbach, 2019; Loose et al., 2020; Kostov et al. 2021; Boland et al. 2021; Kostov et al., 2022), but we are the first to use this framework for reconstructing variability in the volume of LSW accumulated in the Labrador Sea. In addition, our method allows us to attribute historical watermass anomalies to different surface boundary conditions and to identify some of the physical mechanisms that govern the LSW volume budget in the ECCOv4 state estimate (Forget et al., 2015; Fukumori et al., 2017; Heimbach et al., 2019)."

We have now added a citation to Czeschel et al. 2010 (line 132). In lines 403-412, we have expanded our comparison to the results of Jones et al. (2018) and Loose et al. (2020):

"Another important feature in the sensitivity to wind stress is the pattern around the coast of Iceland. We see that a westward wind stress anomaly along the south coast and an eastward wind stress anomaly along the northern coast of Iceland promote a larger LSW volume at a lead time of three years (Figs. 4a, 5a). Jones et al. (2018) identify a similar pattern in the sensitivity of Labrador Sea heat content to wind stress. In addition, Loose et al. (2020) see this pattern in the sensitivity of heat transport across the Iceland-Scotland Ridge to wind stress. Loose et al. (2020) highlight Ekman transport along the Icelandic Coast as the mechanism behind this pattern and argue that it can generate onshore convergence or divergence and hence a pressure anomaly along the coast. This pressure anomaly is quickly communicated around Iceland as a coastal wave and affects the geostrophic transport between Iceland and Scotland, heat transport convergence in the Nordic Seas (Loose et al., 2020), and subsequent water mass transformation. Through Denmark Strait (Fig. 1), temperature and salinity anomalies from the Nordic Seas, are then exported back to the eastern subpolar gyre, where they can precondition LSW formation. "
That is in addition to the already existing comparisons and parallels drawn in lines 397 and 568.

We already stated in lines 569-570: "Our findings are thus similar to the results of Jones et al. (2018), who highlight similar enhanced sensitivity of total heat content in the Labrador Sea to wind stress along the African and West European shelves."

Presumably, the novelty is in the use of the linear adjoint technique for this topic, noting that the adjoint had been used similarly for other applications. Overall, I feel somewhat on the fence about this manuscript.

We have tried to be more explicit about the new contributions that our study presents and put them in context. For example, in lines 136-140 we now say "Our results suggest that the upper limb of the AMOC exerts a strong lagged influence on the rate of LSW formation, a feature seen in some but not all general circulation models (Ortega et al., 2017; Li et al., 2019). It is noteworthy and novel that we identify this causal relationship between the upper AMOC limb and LSW in the ECCO state estimate constrained with historical surface boundary conditions and observations of the real ocean (Forget et al., 2015)."

We have also edited lines 563-564 following your suggestion, and there we once again highlight some of the novelty: "These novel results challenge the traditional view that wintertime cooling in the Labrador Sea is the dominant driver of interannual variability in LSW volume in the Labrador Sea." (Previously that sentence referred to NADW).

We also elaborate on the importance of our contribution in lines 579-585 : "Some general circulation models show a significant lagged correlation between the upper AMOC limb and LSW, where the former leads the latter in time (Ortega et al., 2017; Li et al., 2019). We are the first to identify this as a causal relationship and an oceanic teleconnection in a state estimate constrained with observations (Forget et al., 2015). Our "Traffic Controller" sensitivity pattern is a geographical fingerprint associated with this oceanic teleconnection that relates flow along the upper AMOC limb to LSW formation and storage in the Labrador Sea. Surface wind stress and density anomalies can act to divert and redirect the transport of warm and saline subtropical water which is necessary for the formation of LSW in the Labrador and Irminger Seas."

We furthermore want to clarify that our approach is fundamentally different from previous studies of watermass budgets, and that the difference extends beyond the use of the adjoint. In lines 124-130, we now say, "Our approach is different from the watermass formation and transformation framework of Walin (1982), Speer and Tziperman (1992), and Desbruyeres et al. (2019) who also use surface fluxes in their analysis. Our main constraint for defining LSW is based on vertical stratification (a component of the potential vorticity, PV), while we also define generous potential density bounds to help identify the watermass. This is similar to the definition of LSW used in Li et al. (2019). Another important difference between our framework and Speer and Tziperman (1992) is that we consider the immediate and delayed impacts of both local and remote surface heat and freshwater fluxes, as well as surface wind stress across the entire Atlantic – Arctic region."

It keeps jumping from one point to another with quite a few figure panels, not all of which are discussed in detail. There are many small spatial scale features in the figures and their interpretations are not usually prvided.

Figure 2 is now Figure 3, and we have added further discussion in lines 371-378 following your recommendation and that of Referee 1: "For example, salt fluxes, heat fluxes, and wind stress, all, contribute to the positive LSW volume anomalies in the early 2000s. We suggest that the 2008 relative increase in LSW explored by Yashayaev and Loder (2009) can be attributed primarily to surface heat fluxes (Fig. 3a). These attribution results hold true across our three reconstructions. However, all releases of ECCOv4 seem to underestimate the magnitude of the 2008 relative increase in LSW, where the underestimation is most pronounced in release 4. We find that the subsequent 2010-2011 decline in LSW volume is dominated by heat fluxes and wind stress, while the 2012 recovery is attributable to both heat and salt fluxes (Fig. 3a). The well-documented increase in LSW volume after 2015 (Yashayaev and Loder, 2017) seems to be primarily related to wind stress anomalies and, to a smaller extent, surface heat fluxes."

In addition, following your recommendation, we have described in greater detail the small spatial scale features in Figures 4 and 5 in lines 398-412: "We also see sensitivity to zonal wind stress along the boundaries of the Labrador Sea (Figs. 4a, 5a), which can affect the strength of the boundary current and its exchange with the interior (Czeschel, 2004).We further notice that at a lead time of three years, LSW volume is sensitive to zonal wind stress just south of Greenland (Figs. 4a, 5a) in the region of the Greeland Tip Jet, where zonal winds can affect the rate of deep convection in the Irminger Sea (Pickart et al., 2003b).

Another important feature in the sensitivity to wind stress is the pattern around the coast of Iceland. We see that a westward wind stress anomaly along the south coast and an eastward wind stress anomaly along the northern coast of Iceland promote a larger LSW volume at a lead time of three years (Figs. 4a, 5a). Jones et al. (2018) identify a similar pattern in the sensitivity of Labrador Sea heat content to wind stress. In addition, Loose et al. (2020) see this pattern in the sensitivity of heat transport across the Iceland-Scotland Ridge to wind stress. Loose et al. (2020) highlight Ekman transport along the Icelandic Coast as the mechanism behind this pattern and argue that it can generate onshore convergence or divergence and hence a pressure anomaly along the coast. This pressure anomaly is quickly communicated around Iceland as a coastal wave and affects the geostrophic transport between Iceland and Scotland, heat transport convergence in the Nordic Seas (Loose et al., 2020), and subsequent water mass transformation. Through Denmark Strait (Fig. 1), temperature and salinity anomalies from the Nordic Seas, are then exported back to the eastern subpolar gyre, where they can precondition LSW formation."

I worry about the caveats listed on l.463-473, especially the robustness of the findings – though they are similar to the ones found in other studies.

We wanted to state these caveats openly, and that is why we tried to describe them in detail. At the same time, as you say, putting the results into the context of previous literature and tying them to physical mechanisms gives us further confidence in the robustness and relevance of our findings.

There seem to be larger NAO events (Fig. B1) in 1995, 1996, and 2000. Why weren't those chosen? Can additional runs, including for the summertime, be included, increasing the ensemble size? I think the robustness / fidelity of the "traffic controller" pattern should be quantified.

In lines 233-237, we have now justified the choice of years towards the end of the ECCOv4 release 2 timeseries for defining objective functions: "The selected three years are close to the end of the ECCOv4 release 2 state estimate because that allows us to compute lagged sensitivity

over longer periods leading up to these three years. We use the sensitivity patterns for the ECCOv4 release 2 objective functions to reconstruct not only LSW variability in release 2, but also to attempt reconstructing LSW in the more recent ECCO release 3 (not shown) and in release 4, that extends further in time until the end of 2017."

We had limited computational resources early on and currently no access to a computational budget for this project. We have added two more runs with summertime objective functions (e.g., the new Fig.2, Fig. 3, Figure 11), but we cannot extend the ensemble further by considering objective functions in additional years.

Also, what is the reason for using freshwater fluxes for the forward experiment? Would use of momentum and heat fluxes from Fig. 4 produce similar results?

We have now clarified in lines 432-435: "The spatial pattern of the sensitivity to surface freshwater fluxes is very similar to the sensitivity to heat fluxes (Fig. 4). However, compared to heat content anomalies, freshwater anomalies do not directly trigger air-sea feedback mechanisms that are excluded from our sensitivity patterns. We thus choose to apply the sensitivity pattern from Fig. 4d as a perturbation to the background rainfall…"

The sensitivity to momentum (meridional and zonal windstress) in Fig. 4 has a different geographical fingerprint compared to surface buoyancy fluxes. At the same time, we also see that the NAC pathway stands out in Fig. 5a.

Other comments and suggestions:

l.19-20: This sentence is stronger than what is said in the text on l.409-411.

We have rephrased line 20 to read, "predict a limited yet substantial and significant fraction of LSW variability"

l.23: Rephrase the last part of this sentence.

We have rephrased lines 22-24 to read, "We point out the important role of key processes that promote the formation of LSW both in the Irminger and Labrador Seas: buoyancy loss and preconditioning along the NAC pathway, in the Iceland Basin, the Irminger Sea, and the Nordic Seas."

Fig. 1 caption: Indicate what "R" is both here and elsewhere.

We have indicated that R stands for "Pearson correlation coefficient" when it appears in lines 331, 343, 344, 361, 388, 546, and 553, some of which are figure captions.

l.89-91: Include spatial resolution information.

We added: "The global state estimate has a nominal 1° horizontal resolution and 50 vertical levels." on line 150.

l.106: Delete the comma after "Lozier".

Fixed (Line 173)

l.116: "in Appendix A", that is delete "the".

Fixed (Line 197)

l.121: Use larger parentheses for the most outside ones.

Fixed (Line 203).

l.127: "Figs."

Fixed (Line 208)

l.132-135: I am not following what is said here. How does a simple scaling reduce noise? Also, despite what is said here, the units of 10^14 m^3 seem to be in use throughout the manuscript.

We have added several sentences to clarify this in lines 213-219: "Instead, we multiply the LSW volume by a large nondimensional factor of 1500. The rescaling increases the magnitude of the sensitivity patterns output by the adjoint. This eliminates some (but not all) of the numerical noise that arises when the adjoint of the MITgcm outputs the sensitivity of our objective function to surface boundary conditions. The rescaling helps as some sources of numerical noise in the adjoint have a magnitude independent of the magnitude of the objective function and its sensitivity. We then divide the lagged sensitivity patterns that the adjoint outputs by 1500 when we post-process them offline, so that the post-processed sensitivity has units of $m^3$ per forcing, assuming that the forcing is sustained over a single model time step set to 1 hour."

l.137: "measure" -> "compute".

Fixed (line 221).

l.140-141: How do these seasonal objective functions relate to monthly LSW calculations, say, as plotted in Fig. 1b?

We have added text to clarify this: "That is, we assume that a monthly objective function from March is representative of January and February monthly objective functions, too. Similarly, an August objective function is assumed to be a good substitute for July and September monthly objective functions. In addition to the 2006-2007 set of seasonally representative objective functions, for comparison, we do adjoint calculations with wintertime objective functions computed in 2006 and 2011, and summertime objective functions computed in 2005 and 2010." (Lines 225-230)

l.144: Please refer to much earlier works on this topic.

We have also cited Dickson et al. 1996 and Rhein et al., 2017 (Line 231)

l.154: Subscripts "N" and "E" do not appear to refer to "zonal" and "meridional". Please clarify.

We have now clarified this: "zonal (positive eastward)" and "meridional (positive northward)" (Lines 247, 365, 735) and similarly, in line 381.

l.179-181: How does this impact the adjoint optimization? Does it mean that the surface fluxes are not being adjusted?

The surface forcing remains the adjusted one, just as in the ECCO run. We have now clarified this in line 279: "However, following our approach, the model's forward trajectory remains the same as in the optimized ECCO state estimate."

Figs. 2 & 3: They can be combined to a single 6-panel figure.

Done. See the new combined Fig. 3

l.271: Both here and elsewhere, specify what is meant by "short lead times".

We have changed this to read "at lead times shorter than a year" (lines 391, 393, 529, 729, 730).

l.280: "3c" -> "2c".

The figures and their captions were merged in the new combined Fig. 3, where this typo is no longer present.

l.299: Use "NAO" as it was introduced earlier.

Fixed (Line 420).

l.307: What does "themselves" refer to?

To avoid confusion, we have dropped the word "themselves" in line 423.

l.317: Why is this particular scaling used? Also, why such a small perturbation? Is it because the adjoint has to be linear and it cannot accommodate a larger perturbation?

We have tried to clarify this in lines 437-439: "However, we multiply the pattern by a factor of $(-10^{-22})$, so that the rescaled perturbation is of order $10^{-8} \, m \, s^{-1}$ (or $\sim 10^{-5} \, kg \, m^{-2} \, s^{-1}$), which is comparable to the standard deviation in January surface freshwater fluxes between different years (see Fig. 8b)."

l.317-318: What is the reason for imposing only one-month perturbation and only in January 2000?

We have tried to explain our motivation more clearly in lines 439-445: "Perturbations applied during the winter are distributed over a deeper mixed layer, which further enhances their persistence and triggers a large response. This motivates our choice to branch the experiment from the ECCO state estimate (our control run) in January. On the other hand, the perturbation we prescribe is based on a sensitivity pattern from a winter-time objective function, so it would not be appropriate to extend the prescribed forcing anomaly beyond the winter season. We thus apply the perturbation only throughout January 2000 and explore its impact over the subsequent years. We have chosen the period 2000-2008 because it is marked by a resumption in the formation and storage of relatively larger volumes of LSW (Fig. 2). Hence, launching our experiment in 2000 allows us to explore this regime of enhanced LSW production."

l.320-321: This adjustment is unclear. Why are the poles adjusted, rather than considering an area-average adjustment?

We have explained the motivation behind our choice in greater detail in lines 446-449: "We adjust the amplitude of our positive and negative poles in the applied perturbation pattern such that the net input of freshwater is zero. Adjusting the poles of the applied forcing pattern is more physically consistent than redistributing the freshwater imbalance as a uniform area-averaged offset over the North Atlantic. Unlike a uniform redistribution of the imbalance, the poles of the sensitivity pattern are aligned with dynamical barriers such as inter-gyre boundaries."

l.322-323: I cannot see this "deceleration".

We have clarified this in lines 450-474: "When comparing against the unperturbed ECCO state estimate, we see that the "Traffic Controller" pattern affects the SSH gradients in the subpolar gyre (Fig. 6a). In the climatology (contour lines in Fig. 6a), the SSH decreases in the direction towards the Labrador Sea and Southeast Greenland. In contrast, the SSH anomaly in our experiment is more positive in the Labrador Sea and more negative along the NAC pathway (Fig. 6a). Contours of the SSH anomalies can be used to infer changes in the geostrophic component of surface currents (Jones et al., 2023). It is also important to note that in this freshwater flux experiment, we keep wind stress unperturbed, and hence do not change ageostrophic wind-driven transport relative to the control state. Therefore, the SSH anomalies in Fig. 6a tell us that our "Traffic Controller" perturbation *decelerates* NAC transport towards the western subpolar gyre (blue arrow in Fig. 6a) relative to the control run. However, the "Traffic Controller" increases northeastward transport towards the Iceland Basin and the GIN Seas (red arrow in Fig. 6a) compared to the control simulation. In addition, there is an increase in the southward transport along the East Greenland Shelf (Fig. 6a)."

l.328: delete "water".

Fixed (Line 480).

Fig. 6 caption: "cm" -> "m". Also, "SSS" has not been defined yet.

We corrected the unit typo and defined SSS in the Fig. 6 caption.

l.355 & 357: Both here and elsewhere, no need to repeat "at each model grid point".

We have fixed this in line 497 and the Fig. 8 caption.

l.358 & 359: Fig. 8 is for January, so winter is shown, correct?

We added "in the winter season compared to the summer" (line 506)

l.361: Please update the date for Petit et al. citation, both here and elsewhere.

Updated on lines 508-509; 591; and updated in the references.

l.362: " …. Surface buoyancy flux …. ".

We added the word "buoyancy". (Line 509)

Fig. 9 caption: To be precise, Fig. 4g is for 61 months, not 5 years.

We have changed the Fig. 9 caption, as suggested.

l.386-387: What does "short characteristic lead time" mean?

We have changed that to read "characteristic lead times that peak within two years" (Line 528)

l.391: "drive even more delayed response" why?

We have added a possible explanation: "This is consistent with the findings of Kostov et al. (2022) who suggest that LSW volume responds to surface perturbations along the Labrador Sea Western Boundary on timescales set by the propagation of signals from the western to the eastern subpolar gyre and back." (Lines 533-535)

l.392: Figure 10 panels have large magnitude differences. Please discuss implications. Are all regions equally important?

We have added a discussion in Lines 536-540: "Surface boundary conditions over different regions make contributions of very different magnitudes to intermonthly (Fig. 10) and interannual (Fig. F2) LSW variability. Among the four regions that we focus on, surface boundary conditions along the NAC pathway make the largest contribution (Figs. 10c and F2c). This further highlights the importance of the "Traffic Controller" pattern that we identify and its role in driving LSW volume anomalies via alterations to the strength and the pathway of NAC transport."

Fig. 10: Panel titles do not seem to match what these plots are. They are supposed to be contributions of the surface flux components into these regions. Not the other way around. Also, l.401: "panel b" -> "panel c".

We corrected the Fig. 10 panel titles and caption.

l.405-406: What is the reason for the "1 year" choice?

Our skill decreases drastically if we extend our annual-mean prediction two years into the future (not shown). This sharp decline in prediction skill is not a surprise, as the responses to surface heat and momentum fluxes along the NAC exhibit peaks at lead times of roughly 1.5 to 2.5 years (Fig. 10c and F2c). (Lines 555-558)

Fig. 11: Panel title does not match what is shown.... 1 year into the future, not longer than 1 year.

The Fig. 11 panel title was fixed.

l.425-426: I am not sure if this sentence is correct. The traditional view concerns (multi-)decadal time scales. The present work does not really cover that time horizon.

We have changed the text to read "These novel results challenge the traditional view that wintertime cooling in the Labrador Sea is the dominant driver of interannual variability in LSW volume in the Labrador Sea." (Lines 563-564). Previously, the sentence vaguely referred to "North Atlantic Deepwater (NADW) masses" which inappropriately implied that we were referring to the entire low branch of the upper AMOC cell. Thank you for pointing this out.

l.427: "significant" statistically?

Yes, we added "statistically" (Line 565-566)

l.430: Delete parentheses for Pillar et al.'s year.

Changed (Line 568)

l.439: Delete "be".

Changed (Line 577)

l.456: "plays a key role" -> "contributes".

Changed. (Line 602)

l.469" "on" -> "in".

Fixed (Line 614)

Fig. A1 caption: "in a" -> "is" ?

We rephrased that text (Fig. A1 caption)

l.503: What does "This" refer to?

We have changed the text to read "The steep maximum slope of the activation function raises the question …" (Line 649)

l.526: No need to repeat the figure information again.

Fixed (Line 672)

l.549: Again, why this citation? There are many seminal ones on this topic.

We have also cited Dickson et al. 1996 and Rhein et al., 2017 (Line 695)

l.580: Why does the skill peak around 2.5 years? Why is it rather low early on? Also, the decline of skill is rather small. So, does not really justify the cutoff at 6.5 years.

It is beyond the scope of our study to establish the full set of processes that explain the change in reconstruction skill with time. The reconstruction skill is low for lead times shorter than 2 years because various processes (such as the Traffic Controller mechanism) affect LSW volume at lead times beyond 2 years.

The decline of skill is indeed small after 6.5 years. What is more important is that using a history of forcing longer than 6.5 years does not improve the reconstruction. Instead it adds more noise, even if the skill does not deteriorate.

l.581: "for all points on" -> "in".

Fixed (line 727)

Appendix D: Is this Appendix really needed?

Thank you for this question! When we presented this work at meetings prior to submission there were always questions about the sensitivity at short lead times. That is why we decided that a subset of the readers may want to see these results, and we kept them in the Appendix.

l.604: "Irminger".

Fixed (Line 749)

Fig. F1: Why not just use one panel only with different colors for regions? In the caption: "Labrador".

Following your recommendation, we have merged the different panels in Fig. F1, and individual regions are distinguished using colors. We have also fixed the typo in the Fig. F1 caption.

Fig. F2 caption: Same comments as for Fig. 10 caption above.

We have updated the Fig. F2 panel titles and caption following your recommendation.

l.665: "the helpful" -> "helpful".

In this case "for the helpful discussions" may be correct. (Line 810)

l.678: Delete the first sentence.

We have deleted the first sentence. See the updated paragraph starting on line 823.

l.682: Is this language still acceptable for the journal?
The relevant code developed in this study will be made publicly available

---

## Author Response (AR2)

Dear editor and referees,

As the lead on this manuscript, on behalf of all authors, I would like to thank everyone who has contributed to the reviews. We appreciate and value these diverse perspectives on our work and their helpful suggestions. We have tried to address all remaining comments and thereby improve our manuscript. Please find our detailed responses below.

Kind regards,

Yavor Kostov

On behalf of all authors

Referee #1

The authors have done an excellent job revising the paper, especially in terms of the literature and fitting the work into the context of previous studies. They have alsodone a good job explaining things more clearly. At this point I think the paper is basically suitable for publication. My only comment is I think the paragraph added on theECCO representation of LSW, and the new figure 2, would be better if moved to section 2.1 (where ECCO is discussed), as it feels a little out of place in the introduction.

We thank Referee #1 for the helpful comments. We have moved Figure 2 and the text from the introduction that referred to Figure 2 to lines 147-172 in Section 2.1, as you suggested.

Referee #2

The authors have addressed / incorporated many of the reviewers' comments and suggestions. I think the manuscript is much improved, putting the present work in better context with the previous literature. I have several additional comments and suggestions as discussed below. I recommend acceptance after consideration of the items listed below (minor revisions).
We thank Referee #2 for the helpful reviews.

While some additional information has been provided on ECCO (and ECCOv4), what the differences are between the three releases have not been given, including the integration periods. This information should be provided in Section 2.1.

We have expanded this description in lines 137-147 to elaborate on differences between releases, including integration periods.

Also, please be consistent with the release notations throughout the text and figures, that is, either use, for example ECCOv4 release 4 or ECCOv4r4.

Thank you, we have corrected this to be more consistent in lines 137-147 and throughout the rest of the text.

The analysis considers certain months, but no justification is provided for why those months were chosen. For example, Fig. 4 uses months 37 and 61 without saying why they were chosen.

In line 405, we now say that we consider sensitivities on interannual timescales (e.g., three and five years). Similarly, in lines 476-479 we now say, " We use our freshwater flux perturbation experiment to explore the adjustment of the subpolar North Atlantic and the response of LSW volume on intermonthly and interannual timescales (e.g., one, three, and five years after the surface perturbation)."

Then, another piece of analysis uses month 38 later (Fig. 6). Why? Why not use the same month 37.

For consistency, we have now used Month 37 in Figure 6. This is exactly 3 years after the end of the perturbation sustained throughout Month 1. This is now mentioned in the caption.

Please include justifications for all these choices. Otherwise, it feels like there is some cherry picking going on.

In line 405, we now say that we consider sensitivities on interannual timescales (e.g., three and five years). Similarly, in lines 476-479 we now say, " We use our freshwater flux perturbation experiment to explore the adjustment of the subpolar North Atlantic and the response of LSW volume on intermonthly and interannual timescales (e.g., one, three, and five years after the surface perturbation)."

For consistency, we have now used Month 37 in Figure 6. This is exactly 3 years after the end of the perturbation sustained throughout Month 1. This is now mentioned in the caption.

We have, however, displayed an additional result that does not correspond to a whole number of years after the freshwater flux perturbation. In Appendix E, Figure E1a, we point out that in April, 15 months after the freshwater perturbation, there is an increase in LSW volume in the Irminger Sea. The transient response of the Irminger Sea complicates our picture, but leaving out this result would constitute an oversimplification, which we want to avoid.

The manuscript promotes / implies enhanced predictive skill associated with the NAC pathway in several places. This is summarized on l.595-597. The implication is that there will be high skill perhaps up to pentadal timescales, but skill even for the first year is rather low. In my view, the results do not support what is implied / promoted. These sentences should be accordingly modified.

We gave modified the new lines 614-615 to read "The existence of such large delayed responses in the ocean system implies that surface conditions along the NAC pathway are a major source of LSW variability." (rather than a major source of predictability)

Similarly, in lines 567-568 we say that "The existence of long characteristic lead times in the system motivates us to explore the predictability of LSW." We do not say that sensitivity with long characteristic lead times necessarily gives rise to large predictability. Furthermore, in line 618 we talk about "limited" predictability (as we had already done in line 20 in the abstract.)

Finally, is it possible to link LSW thickness to some sort of overturning transport, say in Sv? The implication in the text is that changes in the LSW thickness will translate into "downstream" transport changes.

Thank you for this suggestion! In this work we focus on the volume of accumulated LSW rather than on the export of LSW as an element of the overturning circulation. The accumulation of a water mass and its export are not trivially related, as shown by Petit et al. (2021), cited here. There can be periods of enhanced production but unchanged export.

Desbruyeres et al. (2019), cited here, are able to relate surface-forced watermass transformation to transport variability. An important part of their analysis is identifying the timescale on which transformation and transport are related. In our case pinpointing the timescale on which LSW accumulation translates into transport variability would merit a separate dedicated study because of its importance.

Other comments and suggestions:

l.53: "Iriminger" -> "Irminger".

Done in line 53.

l.159: Insert "meridional" after "Atlantic".

Done in line 147.

l.214: Insert "arbitrary" after "large".

Done in line 224.

l.219: "per forcing units"?

Done in line 229.

l.220: "depends" -> "depend".

Done in line 230.

l.220: "in" -> "for".

Done in line 230.

l.247: Switch the order of tau_N and tau_E and add ", respectively".

Done in line 257.

l.283: Both here and elsewhere, I think "drive" should be replaced with "contribute to".

Done in line 293 and all instances where we interpret our own results. We have kept the word "drive" when discussing previous literature on driving mechanisms in the introduction.

l.373-375: Both here and perhaps elsewhere, please provide some observational values so that the reader can judge how large the ECCOv4 biases are.

Unfortunately, almost all previous literature that we cite considers only section-based or profile-based LSW thickness results rather than basin-wide volume. Li et al. (2019) do provide a volume estimate that we now refer to in lines 163-165.

l.396: Define SSH here, not later on line 415.

Done in line 408.

l.411: "the Denmark Strait".

Done in line 422.

l.426: No need to relist the figures again here as they are just given one line above.

Fixed in line 437.

l.444: "larger" -> "large".
Fixed in line 455.

l.446: I think it will be good to use a different word than "pole". You are referring to the maximum loadings or maximum / minimum values.

Following your suggestion, we have replaced the phrase "our positive and negative poles in the applied perturbation pattern" with the phrase "our positive and negative values in the applied perturbation pattern" in line 472. However, we cannot use the word "loadings," as our sensitivity patterns are not EOFs, and that may confuse readers. That is why, we have no choice but to say "the poles of the sensitivity pattern are aligned with dynamical barriers such as inter-gyre boundaries" (lines 474-475).

Figure 7 and caption: It looks like the sign convention changed for this figure compared to the others. Please use the same sign convention for all the figures.

Thank you for pointing this out. We have changed the sign convention in Figure 7 for consistency, as you suggested. Also, in order to avoid focusing on arbitrary response times, we have now shown anomalies exactly 12 months after the applied perturbation.

l.475-490: This paragraph discusses the figures in Appendix E very extensively. I suggest moving these figures here.

Thank you, but we have to strike a balance, and the main text already has enough figures.

Also, introduce SST and SSS on l.475 and 476.

Done in lines 493 and 494.

Please connect the sentence on l.477-478 to the contents of the very next sentence.

Thank you, we have now tried to make the logical transition in lines 496-505 smoother, as you suggested.

l.480: "the Denmark Strait".

Done in line 505.

l.499: Delete "at each grid point".

Done in line 533.

Figure 9: Correct the title for panel b to say "surface freshwater flux" in both occurrences.

Done in the title of Fig. 9b

l.525: Specify what is meant by lead and lag.

Done in lines 543-544.

l.580-581: It think it will be good to add something like "Although it has been found in previous literature," to clarify that your claim to be the first pertains to use of a state estimate.

Done in lines 598-599. (We now say "highlighted" in previous literature).

l.728: "is" -> "are".

Done in line 748.

Figure D1: Please make the actual plots larger. As is, there is a lot of empty space between the panels.

Done in Figure D1.

l.743-744: I am not really seeing what is stated here in Fig. 1E. Where is the increase in LSW production in the Irminger Sea in Fig. 1Ea? There is not much in the Irminger Sea. Thickening of the LSW layer near Cape Farewell is rather small in Fig. 1Eb. Please rewrite the discussion for this figure, both here and in the main text.

In Appendix E and the new Figure E1a, we now point out that in April, 15 months after the freshwater perturbation, there is an increase in LSW volume in the Irminger Sea. The transient response of the Irminger Sea complicates our picture. We would have a cleaner set of mechanisms had the Irminger Sea not adjusted. However, leaving out this transient result would constitute an oversimplification, which we want to avoid. At the same time, we do want to strike a balance in terms of the amount of information in the main text. That is why, we have left Figure E1 in the Appendix.

l.780: Delete "defined in Appendix F".

Done in line 800.

l.780: "Labradror" -> "Labrador".

Done in line 800.

l.781: "Greenalnd" -> "Greenland".

Done in line 800.